



# Halon-1301 – further evidence of its performance as an age tracer in New Zealand groundwater

Monique Beyer[1]; Uwe Morgenstern[1]; Rob van der Raaij[1] and Heather Martindale[1]

[1] GNS Science, PO Box 30368, Lower Hutt, New Zealand

*Correspondence to*: Monique Beyer (Monique.Beyer@es.govt.nz)

**Abstract.** We recently discovered a new groundwater age tracer, Halon-1301, which can be used to date groundwater recharged after the 1970s. In a previous study, we showed that Halon-1301 reliably inferred groundwater age at the majority of studied groundwater sites. At those sites, ages inferred from Halon-1301 agreed with those inferred from $SF_6$ and tritium, two reliable widely applied groundwater age tracers. A few samples, however, showed reduced concentrations of Halon-1301, preventing meaningful age interpretation from its concentration. These reduced concentrations were likely a result of degradation or retardation of Halon-1301 in the aquifer. However, we couldn't provide full evidence for this due to the limited number of analysed groundwater samples (18 in total). In this study, we assess the potential of Halon-1301 as a groundwater age tracer for a larger dataset of groundwater samples under specific groundwater conditions, including highly anoxic young groundwater which can significantly degrade Halon-1301, to gain more information on the magnitude of occurrence and the causes of reduced Halon-1301 concentrations.

In this study, we analysed 302 groundwater samples for Halon-1301, $SF_6$, tritium and the CFCs CFC-11, CFC-12 and CFC-113. Comparison of age information inferred from the concentrations of these tracers allows assessment of the performance of Halon-1301 compared to other well established and widely used age tracers. The samples are taken from different groundwater environments in New Zealand and include anoxic and oxic waters with mean residence times ranging from < 2 years to over 150 years (tritium-free).

The majority of assessed samples have reduced or elevated concentrations of CFCs, which makes it impossible to infer a reliable age using the CFCs for these samples. Halon-1301, however, reliably infers ages for CFC-contaminated waters. Three other groundwater samples were found to have elevated $SF_6$ concentrations (contaminated). Again, at these $SF_6$ contaminated sites, ages inferred from Halon-1301 agree with ages inferred from tritium. A few samples (14 sites) exhibit reduced concentrations of Halon-1301, which result in elevated inferred Halon-1301 ages in comparison to those inferred from $SF_6$, tritium and/or CFC-113. Assessment of the groundwater environment at these sites gives further insight into the potential causes of Halon-1301 reduction in groundwater.

Overall, Halon-1301 gives age information that matches ages inferred from $SF_6$ and/or tritium for the majority (97 %) of the assessed groundwater sites. These findings suggest that Halon-1301 is a reasonably reliable groundwater age tracer, and is in



particular significantly more reliable than the CFCs, which may have contamination and degradation problems. Halon-1301 thus has potential to become a useful groundwater age tracer where $SF_6$ and the CFCs are compromised, and where additional independent tracers are needed to constrain complex mixing models.

# 1 Introduction

Groundwater age or residence time is the time water has resided in the subsurface since recharge. The determination of groundwater age can aid understanding and characterization of groundwater resources, because it can provide information on groundwater mixing, flow and recharge rates, and volumes of groundwater (Maloszewski and Zuber, 1982; Morgenstern et al., 2010; Gusyev et al., 2014; Hrachowitz et al., 2016).

Groundwater age can be inferred from environmental tracers, such as $SF_6$ and tritium. The currently used age tracers have

limited application ranges and reliability. For example, $SF_6$ has natural sources (e.g. Bunsenberg and Plummer, 2000, 2008; Stewart and Morgenstern, 2001; Koh et al., 2007), which can interfere with the interpretation of age from its concentration. The commonly used CFCs (CFC-11, -12 and -113) have stagnant input functions over the last 25 years (Bullister, 2011), have anthropogenic point sources (e.g. in industrial and horticultural areas) (e.g. Oster et al., 1996; Stewart and Morgenstern, 2001; Bunsenberg and Plummer, 2008, 2010; Cook et al., 2006) and are known to be degradable in anoxic environments

(e.g. Lesage et al., 1990; Bullister and Lee, 1995; Oster et al., 1996; Shapiro et al., 1997). Ambiguous age interpretations can occur from tritium measurements due to similar rates of radioactive decay and decrease in atmospheric concentration, which leads to similar concentrations of tritium in groundwater recharged at different times. This is particularly true for the northern hemisphere, where concentrations in young groundwater are still elevated due to atmospheric H-bomb testing in the 1960s (Taylor et al., 1992; Morgenstern and Taylor, 2009; Morgenstern et al., 2010). To reduce these limitations of

ambiguity and input uncertainty, multiple tracers should be applied complementarily. New groundwater age tracers and/or new groundwater dating techniques are needed to supplement the existing ones. New complementary age tracer techniques are also necessary to resolve the multi-parameter age distributions for more complex mixing models (Stewart et al., 2016).

We only recently discovered a new groundwater age tracer, namely Halon-1301, which can be used to date groundwater recharged after the 1970s (depending on the limit of detection and mixing model assumptions) (Beyer et al., 2014). Halon-

1301 can be measured simultaneously with $SF_6$, which adds only a little to the cost of analysis. Measurement of Halon-1301 and $SF_6$ in the same water sample allows identification of contact with air during sampling (which can lead to contamination of the water sample with the higher modern atmospheric concentration of both $SF_6$ and Halon-1301), contamination from other sources, or degradation (elevated/reduced concentration of either Halon-1301 or $SF_6$). Comparison of Halon-1301 and $SF_6$ inferred ages to tritium inferred ages also allows assessment of processes in the unsaturated zone and confirmation of

degradation/contamination of one or both Halon-1301 and $SF_6$.

We previously assessed Halon-1301's performance as an age tracer against two relatively reliable established tracers, $SF_6$ and tritium, at 18 sites (Beyer et al., 2015), and found that in the majority of assessed groundwater samples ages inferred





from Halon-1301 agreed with those inferred from $SF_6$ and tritium. None of the samples showed significantly elevated concentration of Halon-1301, despite the presence of CFC contamination from industrial sources. This suggests that Halon-1301 from anthropogenic or geologic sources that could interfere with age interpretations are insignificant in aquifers. In the remaining (anoxic) water samples, reduced concentrations of Halon-1301 were found. Based on these findings, we

concluded that reduced Halon-1301 concentrations were likely caused by degradation of Halon-1301 under anoxic conditions or sorption of Halon-1301 to aquifer material. Both contamination and degradation can result in misleading age estimates.

A recent study by Bartyzel et al. (2016) compared tritium inferred mean residence times (MRTs), using lumped parameter models (LPMs) with piston flow ages inferred from $SF_6$, CFC-12 and Halon-1301 at six sites in Poland. They found Halon-

10 1301 ages agreed well with $SF_6$ ages. None of their samples indicated reduced Halon-1301 concentrations, despite assessing relatively old and anoxic waters. However, we observed reduced Halon-1301 concentrations in old anoxic waters in our previous study. They may not have observed reduced concentrations of Halon-1301 because they assessed only six sites and we previously found reduced concentrations in 29% of the assessed sites. Another explanation is that different groundwater environments were encountered in both their and our previous study.

This study aimed to further assess the performance of Halon-1301 as a groundwater age tracer on a larger dataset and covering a wider range of groundwater conditions than previously assessed. Of particular interest was to confirm the absence of local geologic or anthropogenic sources of Halon-1301 that could cause elevated concentrations in groundwater samples, and to confirm the causes of reduced Halon-1301 concentrations found in anoxic groundwater. We analysed 302 groundwater samples for Halon-1301 and $SF_6$ simultaneously and subsequently inferred ages from their concentrations. The

samples were taken from different groundwater environments in New Zealand. The samples were also analysed and dated with tritium and the CFCs (CFC-11, CFC-12, and CFC-113), with mean residence times (MRTs) ranging from < 2 years to over 150 years (tritium-free water). A large proportion (ca. 30 %) of these samples showed reduced or elevated concentrations of CFCs (in particular CFC-11 and CFC-12), which made it impossible to infer a reliable age using the CFCs in these samples. CFC-113 performed considerable better than CFC-11 and -12, with matching ages to that of tritium for

83 % of the sites. Comparison of groundwater ages inferred from Halon-1301 to those inferred from $SF_6$, tritium and CFC-113, the three most widely applied and reliable age tracers, allowed further assessment of the performance of Halon-130 as an age tracer. In particular, the reliability of Halon-1301 for groundwater dating of CFC-contaminated, CFC-degraded, or $SF_6$-contaminated waters was examined, to gain further insight into the contamination and degradation potential of Halon-1301.



## 2 Method

### 2.1 Sampling and analysis

In this study, we sampled 302 groundwater samples across New Zealand from over 20 aquifers, ranging from highly anoxic to oxic conditions (Figure 1). Not all age tracers were determined at all 302 sites, as summarized in Table 1. To prevent
sampling of stagnant water, the well was flushed with at least three times its volume or until DO and EC stabilized. Tritium was analysed in 1 L water samples, using electrolytic enrichment and liquid scintillation counting (LSC) detailed in Morgenstern and Taylor (2009). For analysis of the gaseous tracers Halon-1301, $SF_6$ and the CFCs, groundwater was sampled under rigorous exclusion of air to avoid contamination of the samples with modern air. For determination of CFC-11, CFC-12 and CFC-113, 125 ml glass bottles with aluminium foil cap liners were used. For determination of Halon-1301
and $SF_6$, 1 L brown borosilicate bottles were used. The sampling methods are detailed in van der Raaij and Beyer (under review). The gas samples were subsequently purged and analysed on a gas chromatograph with attached electron capture detector (GC/ECD). Simultaneous analysis of Halon-1301 and $SF_6$ is detailed in Beyer et al. (2014, 2015). The analytical setup for determination of Halon-1301 and $SF_6$ also allowed the simultaneous determination of CFC-12 (Busenberg and Plummer, 2008; Beyer et al., 2014; Bartyzel et al., 2016). However, an appropriately concentrated standard gas is needed to
establish its calibration curve. CFC-12 concentrations were therefore not determined simultaneously with Halon-1301and $SF_6$ in this study. CFC-12 was analyzed separately, together with CFC-11, CFC-113, Ar and $N_2$, as described in van der Raaij and Beyer (under review).

The amount of gaseous tracers in all groundwater samples was determined by establishing a calibration curve (least square fit, forced through 0/0) with certified air standard at various pressures. We analysed blank samples (only containing $N_2$),
which indicated 0 signal for $SF_6$ and Halon-1301. In addition, the statistical difference between the intercept of the calibration curves for SF6 and Halon-1301 (when not forced through 0/0) were not significant (at 99 % confidence). The intercept of the calibration curve was therefore considered insignificantly different from 0, hence the calibration curve was forced through 0/0 to simplify the calibration procedure and to ensure 0 signal is interpreted as a concentration of 0 (fmol/L, e.g.). This procedure follows the suggestion of Helsel and Hirsch (2002) and Caulcutt and Boddy (1983).
The compositions of the certified air standards for analysis of Halon-1301 and $SF_6$ (supplied by the NOAA in 2014) and for analysis of the CFCs (supplied by the Scripps Institution of Oceanography in 2011) are summarized in

Table 2. A calibration curve was established every day before measurement commenced, since the performance of the GC/ECD can change from day to day, due to fluctuations in the environment (e.g. temperature) or aging of the material (e.g.
column fill). If applicable, the amount of gaseous tracer in the water sample was corrected for headspace and/or excess air (by dissolved Ar and $N_2$ determination described in Heaton and Vogel (1981).

**TABLE 1 here**



**TABLE 2 here**

**FIGURE 1 here**

The equivalent atmospheric molar ratio at time of equilibrium (for groundwater samples at recharge) was determined using
the solubility relationship or Clarke-Glew-Weiss fit (Warner and Weiss, 1985) given in Eq. (1). The solubility fit parameters
for Halon-1301, $SF_6$, CFC-11, CFC-113 and CFC-12 are summarized in Table 3. In contrast to the solubility of the CFCs
and $SF_6$, which have been well studied and directly measured (Bullister et al., 2002; Wilhelm et al., 1977), the solubility
parameters of Halon-1301 have only been estimated by Deeds (2008), using the solubility estimation methods of Meylan and
Howard (1991) and Meylan et al. (1996). Actual solubility measurements of Halon-1301 are not available in the literature
(according to our searches and further backed up by personal communication with Daniel Deeds, 06/03/2015). In our
previous study (Beyer et al., 2015), we used modern (equilibrated tap and river) water to estimate solubility and to validate
the solubility estimates. In this study, we confirmed our previous estimate by using solubility estimated from four additional
modern (river) water samples.

$$\ln K_x = A + B \frac{100}{T} + C \ln \frac{T}{100} \qquad (1)$$

with $K_x$ as solubility; estimated as one of Henry's ($K_H$), Bunsen ($K_B$) or Ostwald ($K_O$) coefficient, $T$ as recharge temperature
(in K) and *A, B, C* as solubility fit parameters, given in Table 2. A salinity term can be added to Eq. (1), but this is negligible
for most groundwater applications and so is ignored here.

**TABLE 3 here**

To determine analytical uncertainty, the EURACHEM/CITAC Guide CG4 (Ellison and Williams, 2012) was followed.
Analytical uncertainty included the following uncertainties related to:

- The least square regression (calibration curve),
- The standard gas concentration and recharge temperature,
- Repeatability error from relative standard deviation of replicates, and
- Correction for headspace and excess air.

Uncertainty related to solubility is unknown or has never been reported, so it was not considered in this study. Uncertainty of
the solubility of Halon-1301 is relatively high, approx. 10 % (Beyer et al., 2015), and therefore would add 10 % to the total
analytical uncertainty for the determination of Halon-1301. We believe that Halon-1301's solubility will be determined with
sufficient precision and become available in the near future. To enable comparison of Halon-1301's performance as an age
tracer compared to other tracers after availability of a sufficiently accurate solubility value, we did not include the currently



high uncertainty in its solubility in the following analysis. For the interested reader, the effect of the uncertainty on the age estimate when adding 10 % uncertainty for solubility is shown in Beyer (2015).

## 2.2 Inferring groundwater ages

To infer the recharge year or residence time of the groundwater, the equivalent concentration of tritium, Halon-1301, $SF_6$ and

the CFCs in the atmosphere at time of recharge (determined as described above) was compared to their historic atmospheric records (illustrated in

Figure 2). For tritium, radioactive decay also is applied, with its half-life of 12.32 years. Southern hemisphere atmospheric $SF_6$, CFC-12, CFC-113 and CFC-11 records are available at the GMD/NOAA (http://www.esrl.noaa.gov/gmd/; Thompson et al., 2004) and CDIAC websites (Miller et al., 2008); data from 1973–1995 have been reconstructed by Maiss and

Brenninkmeijer (1998). Southern hemisphere (Cape Grim) atmospheric Halon-1301 concentrations have been summarized and smoothed by Newland et al. (2013). Data from 1969 to 1977 have been reconstructed by Butler et al. (1999). Tritium records for New Zealand are available at Kaitoke, New Zealand. Since seasonal variability of groundwater recharge can affect tritium recharge to groundwater, the tritium recharge is often estimated using recharge weighting techniques (Allison and Hughes, 1978; Stewart and Taylor, 1981; Engesgaard et al., 1996; Knott and Olipio, 2001). Morgenstern et al. (2010)

showed that this is less of a problem in New Zealand, because infiltration is relatively constant through the seasons and the summer gap in infiltration occurs at average tritium concentration in rain, so there is little bias. We therefore did not weight the tritium input in this study. However, the tritium input function was scaled according to elevation and altitude (Morgenstern et al., 2010; Stewart and Morgenstern, 2016).

To account for mixing of waters of different age in the aquifer or during sampling, lumped parameter models (LPMs) were

used (Maloszewski and Zuber, 1982). The use of LPMs allows inference of an age distribution rather than the mean or apparent age of a groundwater sample. The age distribution is increasingly used as an indicator for quality and contamination risks (e.g. the New Zealand drinking water standard (Ministry of Health, 2008) and the European Water Framework Directive (EU Legislature, 2000)). Since we did not have reliable estimates of the best-fitting LPM, we initially used a range of LPMs to tests the tracers' performance to infer age. Specifically, the exponential piston flow model (EPM), the dispersion

model (DM) and the partial exponential model (PEM) were used (Eq. (2) to (4)). However, since the performance of the age tracers was very similar for the different LPMs employed, we only present results with regard to the EPM.

$$\text{EPM: for } ' > MRT\left(1 - \frac{1}{n}\right), f_{EPM} = \frac{n}{MRT} * exp\left(-n * \frac{t'}{MRT} + n - 1\right); \text{else } f_{EPM} = 0. \quad (2)$$

with MRT = the mean residence time; n = the reciprocal of the ratio of exponential in total flow, which we refer to as E/PM = 1/n, the ratio of exponential to total flow in the following (n has been defined as ratio of total to exponential flow

after Maloszewski and Zuber, 1982). At E/PM = 0 pure piston flow is obtained, and at E/PM = 1 pure exponential flow is obtained. The EPM matches well tritium time series data and therefore is the most commonly used LPM in New Zealand (Morgenstern and Daughney, 2012).



DM: $f_{DM} = \frac{1}{MRT} \times \frac{1}{\sqrt{4\pi DP \frac{t'}{MRT}}} \times e^{-\frac{\left(1-\frac{t'}{MRT}\right)^2}{4DP\frac{t'}{MRT}}}$ (3)

The DM conceptualizes one-dimensional advection-dispersion, with DP as the dispersion parameter, which is defined as $DP = \frac{D}{Vx}$ with D as the dispersion coefficient, V as velocity and x as outlet position. When DP = 0, piston flow behaviour is obtained.

5    PEM: for $t' > MRT_{aq} * \ln(m)$, $f_{PEM} = \frac{m}{MRT_{aq}} * exp(\frac{-t'}{MRT_{aq}})$ ; else $f_{EPM} = 0$. (4)

with $MRT_{aq} = \frac{MRT_s}{\ln(m)+1}$, $MRT_s$ is the MRT of the sample and m is the reciprocal of the ratio of sampled to total volume (P/EM). This version of the PEM conceptualises mixing of water in an aquifer that can be described by the exponential model (EM) with only part of the well being screened/sampled. At n = 1 (for wells screened across the entire aquifer) the EM is obtained.

**FIGURE 2 here**

To quantify uncertainty in the inferred LPM parameters as a result of uncertainties in the determination of the tracers in groundwater, age modelling was placed into a probabilistic framework illustrated in Figure 3. The framework included the

generation of tracer concentrations by random sampling of the model inputs from within their uncertainty. LPMs which generated tracer concentrations within +/- 1 SD of observations were considered as behavioural, i.e. adequately fitting and representative. The remaining LPMs were considered as non-behavioural and were disregarded. Consequently, and in contrast to the commonly inferred single LPM parameter point estimate, age information in this study is determined as behavioural LPM parameter populations (i.e. clouds of MRTs and mixing parameter pairs that produce tracer concentrations

within +/- 1 SD of the observation) illustrated in

Figure 4.

For this study, we considered only uncertainty in the determination of the tracers (i.e. analytical uncertainty, determined as described above). This commonly used approach may underestimate the uncertainty in the age interpretation, but gives a first insight into the performance of Halon-1301 as an age tracer compared to other, better established age tracers. For a more

comprehensive analysis, all model uncertainties, such as the uncertainty in the tracer's recharge estimate, as well as assessment of the appropriateness model components, need to be included in the uncertainty modelling approach, as demonstrated in Beyer et al. (under review) and Beyer (2015); Green et al. (2014), Massoudieh et al. (2014, 2012) and Timbe et al. (2013).

**FIGURE 3 here**



## 2.3 Comparison of tracer performance

Figure 4 illustrates examples of behavioural age interpretations (i.e. population or cloud of LPM parameters that produce

tracer concentrations within +/- 1 SD of the observation) for three sites determined with two tracers. To assess whether Halon-1301 gives comparable age estimates to the ones inferred from $SF_6$, CFC-12, CFC-11 and tritium, we determined if the inferred LPM parameters populations overlapped (i.e. agreed). If they were non-overlapping (i.e. different), we determined the distance of inferred LPM parameter populations as a measure of difference. As a measure of distance, we determined the nearest neighbour and minimum Euclidian distance between two data clouds in Matlab software (Muja and

Lowe, 2009). From that, the % difference in MRT and mixing parameter inferred with two tracers (e.g. $SF_6$ and Halon-1301) was determined.

**FIGURE 4 here**

We decided not use the widely applied one-dimensional comparison of MRTs inferred from different tracers (i.e. MRT(tracer1) versus MRT(tracer 2) plots), since this type of comparison may result in misleading interpretations of the agreement/disagreement between age information inferred from the different tracers. For example, for site C illustrated in Figure 4, one may conclude that both tracers' inferred MRTs agree. However, when assessing the behavioural MRT and mixing parameter population in Fig. 4, it is evident that both tracers' age interpretations do in fact not agree (i.e. the LPM

parameter clouds do not overlap), although the tracers give similar MRT estimates.

## 3 Results

### 3.1 Solubility

The estimated solubility of Halon-1301 using modern equilibrated water samples in this study was comparable to the solubility estimated previously (Beyer et al., 2015) (

Figure 5). We therefore considered the use of the previously estimated solubility coefficients as reasonable for estimating equivalent atmospheric mixing ratios from concentrations of Halon-1301 in water (procedure described in method section). To more accurately determine the solubility of Halon-1301 and reduce uncertainty in its determination, further study is needed (also pointed out in Beyer et al. (2015)). Accurate measurement of the solubility of Halon-1301 is beyond the scope of this study, as due to its extremely low solubility, specialised equipment is required.

**FIGURE 5 here**





## 3.2 Age interpretation

In the following, we discuss age interpretations when employing the EPM only. We note that although the EPM is the most commonly employed LPM in New Zealand and other places around the world, we could not with confidence exclude that groundwater mixing at the studied sites is better represented by the DM, PEM or other more complex models, because time series age tracer data is lacking for the vast majority of assessed sites. However, very similar conclusions in terms of the tracer's age interpretation and agreement/disagreement of the inferred age information can be drawn when using the DM and PEM.

Halon-1301 data were limited to one measurement at each site (Halon-1301 has only recently been discovered). Time series $SF_6$, tritium and CFC data, although available for a few sites, were not employed in this study. We decided to use only one observation at each site for each tracer to infer age, allowing an unbiased comparison of the tracers' performance as age tracers. As a result, relatively large uncertainties in inferred age information were obtained, as shown subsequently. To constrain the uncertainty in inferred age information further and assess the value of time series Halon-1301 data, we are aiming to collect and analyse time series Halon-1301 data in New Zealand groundwater for a direct comparison of time series Halon-1301 and other tracer data.

We emphasize that one should not conclude that the assessed age tracers are not useful because the uncertainties in the age interpretations presented in this study appear large. Instead, this study reiterates what is increasingly recognized in the literature – that there may be issues related to uncertainty in the age estimate and that one needs to apply multiple tracers or time series tracer data to better constrain age information. Further, this study does not attempt to discuss which tracer has the lowest uncertainty in its age interpretation, as this is a complex matter. The uncertainty in the tracer's age estimate is dependent on multiple factors; these include the conditions the tracers' input function (this is dependent on the location of the site), groundwater age and mixing (i.e. the age distribution) at the particular site, in addition to sampling conditions and uncertainty in the determination of the tracers in groundwater. This study only compares Halon-1301 ages with other tracer ages.

First we compared inferred CFC ages with those inferred from tritium to identify the CFC that gives the most reliable age estimates. Misleading CFC age estimates are a common problem (Shapiro et al., 1997; Bartyzel et al., 2016; Stewart and Morgenstern, 2001), because CFCs are prone to degradation and contamination. Thereafter, we assessed the performance of Halon-1301 as an age tracer relative to tritium, $SF_6$ (assuming that $SF_6$ and tritium give the most reliable age estimates) and with the CFC that has been found 'most reliable' in this study (CFC-113). We use the term inferred 'age' as a synonym for inferred 'age interpretation', referring to the cloud of behavioural EPM parameters (

Figure 4).

Figure 6 and Figure 7 illustrate the performance of the CFCs as age tracers relative to tritium (refer to figures in Appendix A for details). Figure 8 illustrates typical LPM parameter populations that were inferred based on the CFCs and tritium. For the majority of assessed groundwater samples, the CFCs gave similar age estimates to tritium. However, 29 % and 38 % of the



sites were contaminated or degraded in CFC-12 or CFC-11, respectively, which made it impossible to reliably infer age from the CFCs at these sites (Figure 6). CFC-113 performed considerably better than the other CFCs (Figure 7) and is therefore considered the most reliable age tracer of the three CFCs in this study.

**FIGURE 6 here**

**FIGURE 7 here**

**FIGURE 8 here**

Figure 9 confirms that $SF_6$ is more reliable than the CFCs. At 94 % of the sites, where both tritium and $SF_6$ data were
10 available, $SF_6$ and tritium MRTs matched. Only six sites were contaminated with $SF_6$. $SF_6$ concentrations of these samples were at least 15 % higher, but in some cases several hundred % above current-day atmospheric concentrations. For these samples, comparison of $SF_6$ and Halon-1301 inferred MRTs was not possible. At all except one of these sites, matching Halon-1301 and tritium inferred MRTs were found. At that one $SF_6$ contaminated site where the Halon-1301 and tritium inferred MRTs did not match, evaluation of tritium data was inconclusive, as it gave an ambiguous age interpretation
(suggesting the water could be either very young (<2 years) or older (>50 years)). That site was Halon-1301 free (and also free of CFC-11 and CFC-12), suggesting the water is older than ca. 75 years. Further, CFC-113 concentrations at this site were very low, suggesting that the water is older than 50 years, which does align with CFC-12, -11 and Halon-1301 data suggesting this site may not be degraded in Halon-1301, although we cannot exclude that the CFCs may be degraded too (this is an anoxic site).

**FIGURE 9 here**

In the following, we further discuss the performance of Halon-1301 as an age tracer in comparison to $SF_6$, tritium, and CFC-113. Of particular interest was the magnitude of occurrence of reduced Halon-1301 concentrations and their cause. In our
previous study, we concluded that the most likely reasons for reduced Halon-1301 concentrations are degradation and sorption of Halon-1301 to aquifer material (Beyer et al., 2015). To further assess the reasons for Halon-1301 reduction in groundwater in this study, we studied the groundwater environment and compared the performance of Halon-1301 with that of the CFCs more closely in samples that indicated presence of reduced concentrations of Halon-1301.

Figure 10 illustrates typical LPM parameter populations that were inferred from $SF_6$, Halon-1301, CFC-113 and tritium data
in this study (figures for all sites are presented in Appendix A). As mentioned previously, we found that the inferred LPM parameter populations were relatively large for most sites, and most tracers, particularly the mixing parameter (E/PM), was difficult to constrain. This is mostly due to the use of only one tracer observation to infer age at each site, which cannot sufficiently constrain the uncertainties in the LPM parameters. To reduce the uncertainty in the inferred LPM parameters further, time series tracer data are needed and/or multiple tracers need to be applied complementarily.



**FIGURE 10 here**

Overall, none of the samples indicated significantly elevated Halon-1301 concentrations (i.e. > 10 % reduced Halon-1301
inferred ages, or concentrations above 10 % of modern-day air). This finding is in line with our previous findings and
suggests the absence of local sources of Halon-1301 that could lead to contamination of Halon-1301 in groundwater.
Considering that these sites cover a large fraction of New Zealand's groundwater systems, Halon appears to not be impacted
by geologic or anthropogenic local sources in general.

Figure 11 illustrates that 99 % of the sites where both tritium and Halon-1301 have been determined showed matching
tritium and Halon-1301 inferred ages (+/-10 %). Since tritium is seen as one of the most reliable age tracers, this finding is
really positive, suggesting Halon-1301 is equally as reliable as tritium at 230 sites, although at 90 sites significant difference
in E/PM (>10 %) were found (Figure 11), which can be related to both tracers' input functions. Specifically, as tritium input
is a pulse function, it is easier to constrain the mixing model (mixing parameter). Halon-1301, however, has an S-shaped
input, making it more difficult to constrain the mixing model (mixing parameter). We note that with $SF_6$, with its near linear
input, we expect that constraining of the mixing parameter is even poorer than with Halon-1301, which is assessed
subsequently.

**FIGURE 11 here**

At 79 % of the sites where comparison between Halon-1301 and $SF_6$ inferred ages was possible (i.e. where both $SF_6$ and
Halon-1301 were determined and $SF_6$ was not contaminated), inferred Halon-1301 MRTs agreed within +/-10 % of the $SF_6$
inferred MRTs (Figure 12). E/PMs inferred from Halon-1301 also agreed with the ones inferred from $SF_6$ at the majority of
sites. Further, Figure 10 (and Appendix A) suggest that in most cases Halon-1301 can constrain the mixing parameter and
the MRT better than $SF_6$, as indicated by the generally 'narrower' LPM parameter ranges inferred with Halon-1301, although
the difference was not statistically significant. We note that in any case, time series tracer data are necessary to better
constrain the mixing parameter and allow a more conclusive comparison of the tracers' performance as age tracers.

For the remaining 21 % of the samples (where comparison of both $SF_6$ and Halon-1301 was possible) Halon-1301-inferred
ages were elevated compared to the ages inferred from $SF_6$. In the following, we assess whether Halon-1301 is likely to be
reduced at these sites (summary shown in Table 4).

At 39 of these 62 sites where Halon-1301 inferred ages were elevated compared to $SF_6$, tritium inferred ages agreed with the
Halon-1301 inferred ages (+/-10 %). This suggests that concentrations of Halon-1301 in these samples were not reduced,
despite elevated Halon-1301 inferred MRTs compared to those inferred using $SF_6$. For only one of the remaining sites was
the Halon-1301 inferred MRT also elevated compared to the MRT inferred from tritium, suggesting reduced concentrations
of Halon-1301 were present in this sample.





**FIGURE 12 here**

The remaining 22 sites at which Halon-1301 inferred MRTs were higher than those inferred using $SF_6$ did not have tritium information. CFC-113 concentrations at only two of these sites agreed with Halon-1301 inferred MRT, suggesting that at these sites Halon-1301 reliably infers age. At seven of the remaining 21 sites, CFC-113 data were unavailable. At five of these seven sites, Halon-1301 inferred MRTs agreed with those inferred with CFC-12 (and at one site also with CFC-11). Although all of these five samples were anoxic and CFCs are also known to degrade under anoxic conditions, the fact that age interpretations inferred from Halon-1301 or CFC-12 data match at these sites suggests that these sites are not likely degraded in Halon-1301 (or CFC-12), but further data is needed to confirm this exclusively.

At the remaining two of the seven sites, both CFC-12 and CFC-11 data were unavailable (in addition to unavailable tritium and CFC-113 data). As we do not have further data, we cannot exclude that Halon-1301 concentrations are reduced in these samples. Anoxic conditions of these samples could suggest degradation of Halon-1301 occurred at these sites, but further data is needed to confirm this supposition.

In summary, this leaves 14 sites with likely reduced concentrations of Halon-1301 (as per comparison to tritium and CFC-113), of which 10 are anoxic, two have an unknown redox state, and one is oxic. At the one oxic site, it is unlikely that reduced concentrations of Halon-1301 were caused by degradation, suggesting that degradation may not (at least solely) be causing a reduction in Halon-1301 concentrations. Sorption in the aquifer remains a possible reason, also suggested by the relatively high MRT (12 years for Halon-1301 and 25 years for $SF_6$). However, mixing of anoxic and oxic water at this site may have occurred, or the redox state may have been otherwise wrongly evaluated. Another possible cause of reduced concentrations at this one oxic site is uncertainties in the solubility of Halon-1301 and uncertainties in the input of Halon-1301 (due to its relatively flat atmospheric concentration over the last 5 years). At the remaining anoxic sites (10 sites or 3 % of the assessed sites), the lack of oxygen strongly suggests degradation of Halon-1301 that can only occur under anoxic conditions.

**TABLE 4 here**

In summary, our findings suggest that Halon-1301 performed well as an age tracer at the majority of groundwater sites. Although reduced Halon-1301 concentrations were found in a few samples, resulting in misleading Halon-1301 inferred age estimates, overall Halon-1301 performed significantly better than the CFCs, which are prone to degradation and contamination (shown in this and our previous study). Figure 13 summarizes the performance of the tracers used in this study, highlighting that Halon-1301 performs almost as well as $SF_6$ and tritium, and with a much higher success rate than the CFCs in this study. In particular, Halon-1301 is significantly more reliable than the CFCs (which were either degraded or



contaminated at as many as 30 % of the sites) and in some cases SF$_6$ (for six contaminated SF$_6$ samples, Halon-1301 still matched age estimates from tritium).

**FIGURE 13 here**

## 4 Conclusion

In summary, this study presented an extensive assessment of the performance of Halon-1301 in 302 groundwater samples across New Zealand. We showed that Halon-1301 had a high reliability as an age tracer, similar to that of SF$_6$ and tritium. It performed much better than the CFCs, CFC-11 and -12, which are prone to degradation and contamination. Both degradation

and contamination lead to non-conforming age estimates. For example, despite some groundwater samples showing evidence of contamination from industrial or agricultural sources (inferred by elevated CFC concentrations), no sample showed significantly elevated concentration of Halon-1301, which suggests there were no local anthropogenic or geologic sources of Halon-1301 contamination.

Like any other tracer, the use of Halon-1301 as a groundwater age tracer has its limitations. In this and our previous study,

reduced concentrations of Halon-1301 were found. Causes for these are likely degradation and/or sorption. Although we provided further evidence for degradation being the main reason for reduced Halon-1301 concentrations, we could not fully determine the reasons for the reduced concentrations. We hope that future studies will explore this matter further. Knowing the cause of reduced Halon-1301 concentrations is important as it can help predict its reliability as an age tracer in different groundwater environments.

Further study is also needed on time series Halon-1301 data to better understand how uncertainty in inferred age information can be constrained with multiple Halon-1301 data compared to other tracers, e.g. tritium and SF$_6$. In addition, the solubility of Halon-1301 needs to be better estimated to reduce uncertainty in the determination of Halon-1301 in groundwater and inferred age.

Overall, we highly recommend the use of Halon-1301 as an age tracer, in particular its use in combination with SF$_6$. The

25 simultaneous determination of Halon-1301 with SF$_6$ (and CFC-12) at no additional cost to sole SF$_6$ analysis, can reduce both tracer's limitations to ultimately obtain a more reliable inferred age than through the use of a single age tracer.

### Acknowledgments

The New Zealand Ministry of Business and Innovation is thanked for funding in line with the Smart Aquifer Characterization (SAC) project.





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

## TABLES

**Table 1. Summary of number of sites at which each age tracer has been determined in this study.**

| Age tracer | # of sites with available data |
|---|---|
| Halon-1301 | 302 |
| SF$_6$ | 302 |
| tritium | 229 |
| CFC-11 | 297 |
| CFC-12 | 297 |
| CFC-113 | 288 |

**Table 2. Concentrations of compounds in calibrated air standard and custom made standard gas in parts per trillion by volume (ppt) SIO-2005 scale.            * estimated from atmospheric concentration at time of filling.**

| Compound | Calibrated air (Scripps) 2011 | Custom made standard gas (NOAA) |
|---|---|---|
| SF$_6$ | 7.53 (±0.81) ppt | 10.97 (±0.04) ppt (analysed) |
| CFC-13 | approx. 5.3 ppt (not reported*) | None |
| Halon 1301 | 3.27 (±1.55) ppt | 29.3 (±0.2) ppt (gravimetric blend) |
| SF$_5$CF$_3$ | approx. 0.16ppt (not reported*) | 18.6 (±0.2) ppt (gravimetric blend) |
| CFC-12 | 530.8 (± 0.06) ppt | 511.4 (±2.0) ppt (analysed) |
| CFC-11 | 238.43 (±0.06) ppt | None |
| CFC-113 | 74.88 (±0.11) ppt | None |
| others | Other CFCs and Halon gases usually contained in air | Halon-1201 (6.31 ± 0.03) |



**Table 3. Reported solubility parameters for Halon-1301 and SF₆ and * solubility parameters for Halon-1301 estimated in Beyer et al. (2015) with an uncertainty of 10 %**

| Compound | Reference | Parameters for Henry solubility coefficient [mol/L/atm] | | |
|---|---|---|---|---|
| | | A | B | C |
| SF₆ | Bullister et al., 2002 | -96.5975 | 139.883 | 37.8193 |
| CFC-11 | Warner & Weiss, 1985 | -134.1536 | 203.2156 | 56.2320 |
| CFC-12 | | -122.3246 | 182.5306 | 50.5898 |
| CFC-113 | Bu and Warner, 1995 | -134.243 | 203.898 | 54.9583 |
| Halon-1301 | Deeds, 2008 | -92.9683 | 140.1702 | 36.3776 |
| | Beyer et al., 2015* | -91.878 | 139.001 | 35.478 |

**Table 4. Breakdown of 62 sites that indicated elevated MRTs inferred with Halon-1301 compared to those inferred with SF₆, and assessment of possibility that the differences in SF6 and Halon-1301 inferred MRTs have been caused by degradation of Halon-1301 in groundwater**

| Agreement of Halon-1301 inferred MRT with those inferred with CFCs and tritium | Redox state | # of sites affected | Halon-1301 likely degraded? |
|---|---|---|---|
| Matching tritium and Halon-1301 inferred MRTs | Various | 39 | No |
| Matching CFC-12 and Halon-1301 inferred MRTs, unavailable CFC-113 and tritium data | Various | 5 | No |
| Matching CFC-113 and Halon-1301 inferred MRT, unavailable tritium data | Various | 2 | No |
| CFC-113 and Halon-1301 MRTs do not match, unavailable tritium data | Oxic | 1 | No, Halon-1301 possibly retarded |
| | Unknown | 2 | **Cannot be excluded** |
| | Anoxic | 10 | **Yes** |
| Tritium and Halon-1301 MRTs do not match | Anoxic | 1 | **Yes** |
| unavailable CFC and tritium data | Anoxic | 2 | **Cannot be excluded** |





# FIGURES

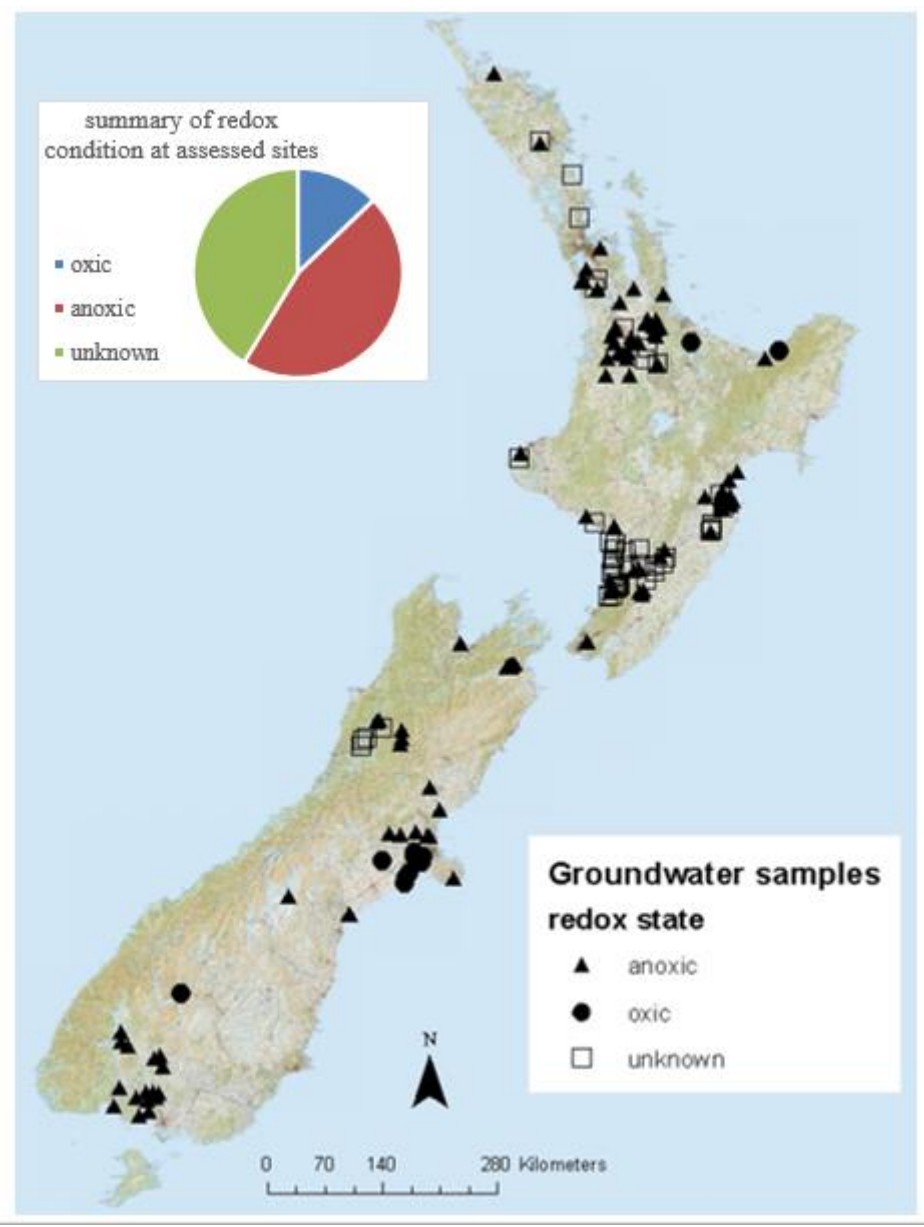

**Figure 1: Location of groundwater samples analysed for Halon-1301 in New Zealand. Groundwater was considered as oxic if the concentration of dissolved oxygen exceeded 0.5 mg/L and/or the concentration of dissolved iron and/or manganese was below 0.05 mg/L and methane was not present (and vice versa for anoxic water).**

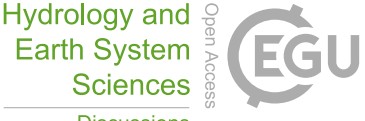



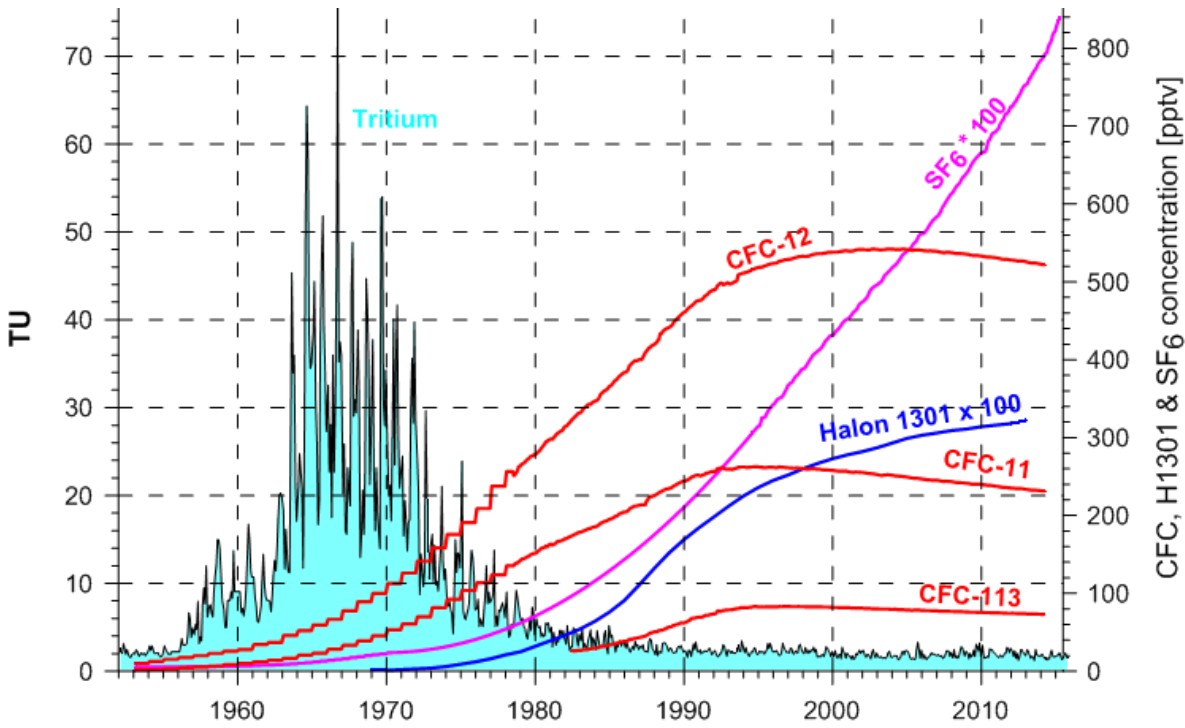

**Figure 2: Southern hemisphere atmospheric concentrations of CFC-12, CFC-11, CFC-113, SF₆, Halon-1301 and tritium, using data from NOAA (available at ftp://ftp.cmdl.noaa.gov/hats) for the CFCs and SF₆; Morgenstern and Taylor (2009) for tritium.**

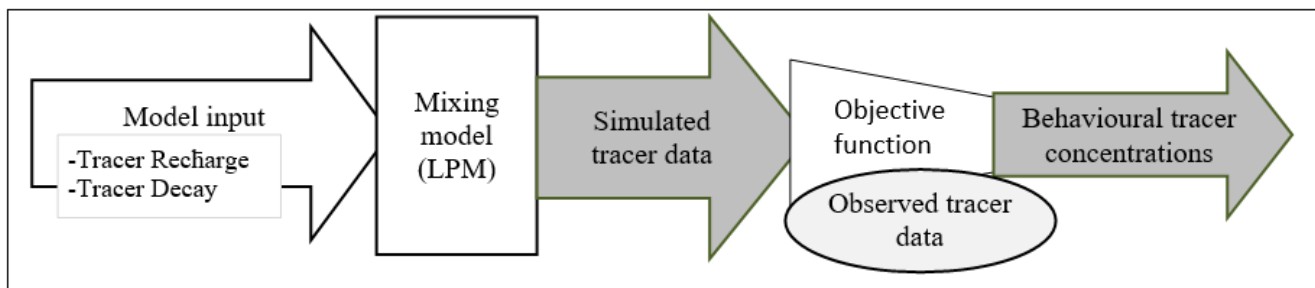

**Figure 3: Schema of the modelling approach with framework components: model input, mixing model and objective function**





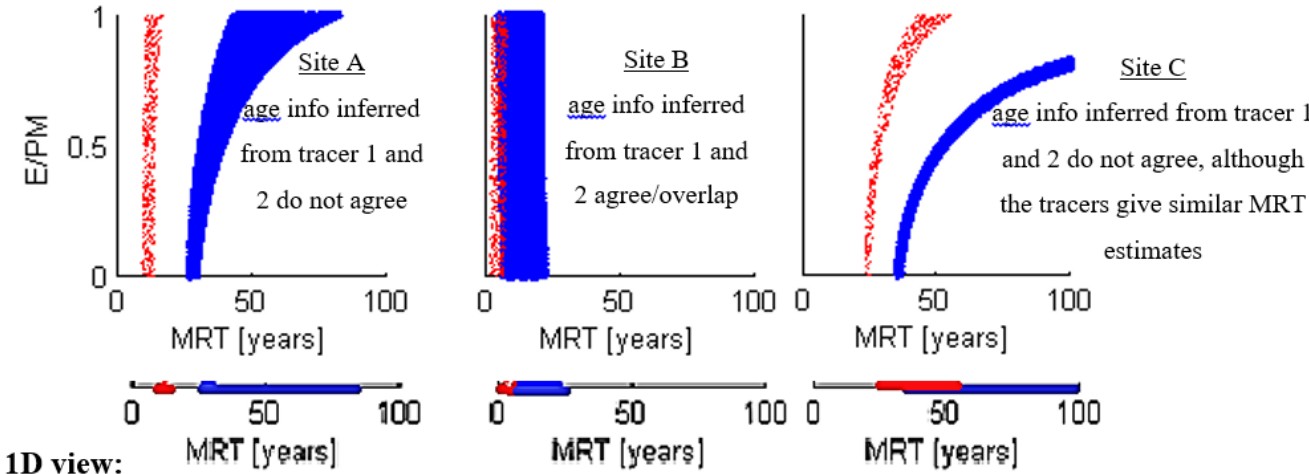

**Figure 4: Example of behavioural lumped parameter model (LPM) parameter clouds inferred from two different tracers (tracer 1 in red and tracer 2 in blue). For site B (MIDDLE) the inferred LPM parameter clouds overlap, indicating that the age**
**interpretation inferred from each tracer agree. For site A and C (left and right) the inferred LPM parameter clouds do not overlap, indicating that the two tracers give different age interpretations. For site C the tracers give similar MRT estimates giving the impression that the tracers age info agree when only looking at one dimension (bottom of each figure). To quantify difference/disagreement, the distance between the data clouds can be determined. [it might be useful to have one more example where the age info 1 and 2 overlap only at a certain E/PM ratio – this is a very realistic scenario using two tracers to constrain the**
**E/PM ratio]**

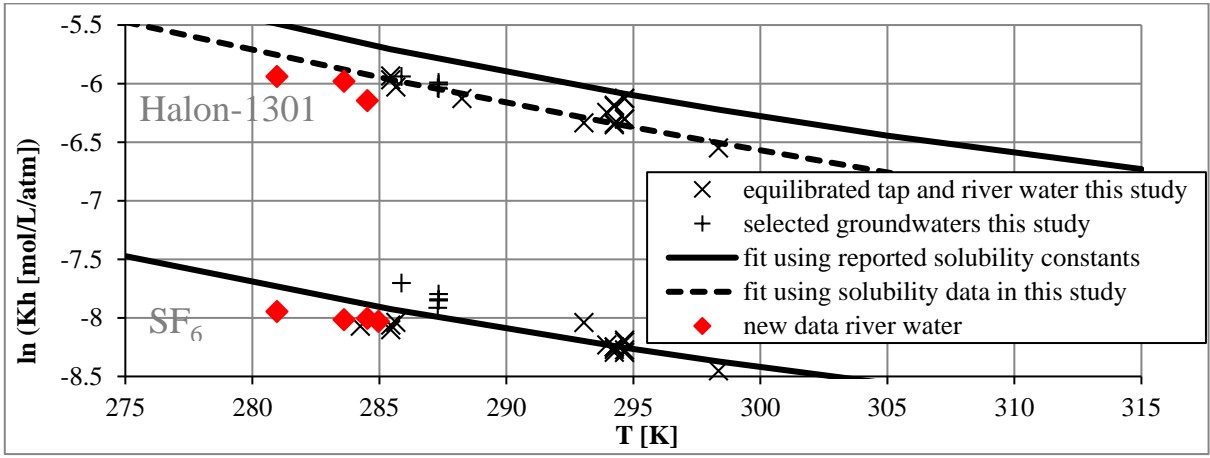

**Figure 5: estimated solubility of Halon-1301 and SF₆ in equilibrated tap water, river water, and oxic young groundwater in comparison to reported solubility data, * data from Deeds (2008) for Halon-1301 and Bullister et al. (2011) for SF₆**





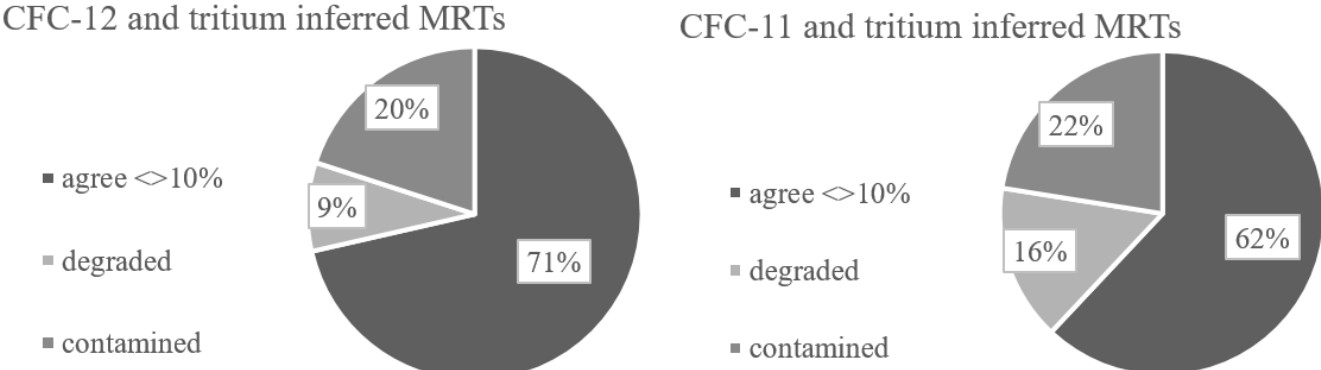

**Figure 6: Agreement of CFC-12 and tritium (right), CFC-11 and tritium (left) inferred MRTs. The data suggest that for most sites CFC-11/ CFC-12 inferred MRTs match tritium inferred MRTs. However, over 30 % of the sites are contaminated or degraded in**
5 **CFC-11/CFC-12, common causes for erroneous age information with CFC-11/CFC-12.**

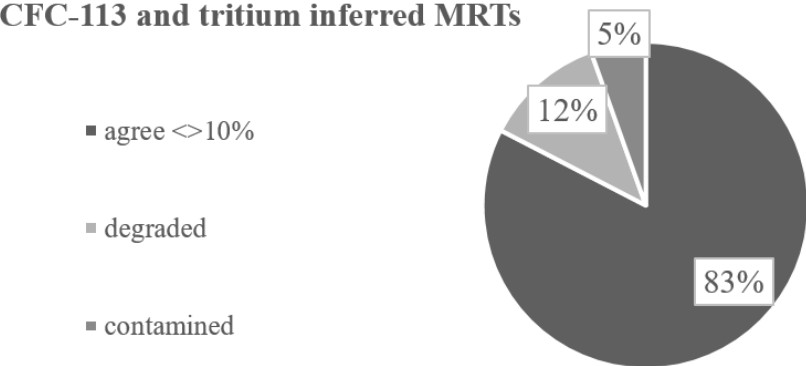

**Figure 7: Agreement of CFC-113 inferred MRTs compared to those inferred with tritium, suggesting that for most sites MRTs inferred from CFC-13 match tritium inferred MRTs. Some (5 %) of the sites are contaminated with CFC-113 or (12 %) degraded**
10 **in CFC-113. Overall CFC-113 performs considerably better than CFC-11 or CFC-12.**



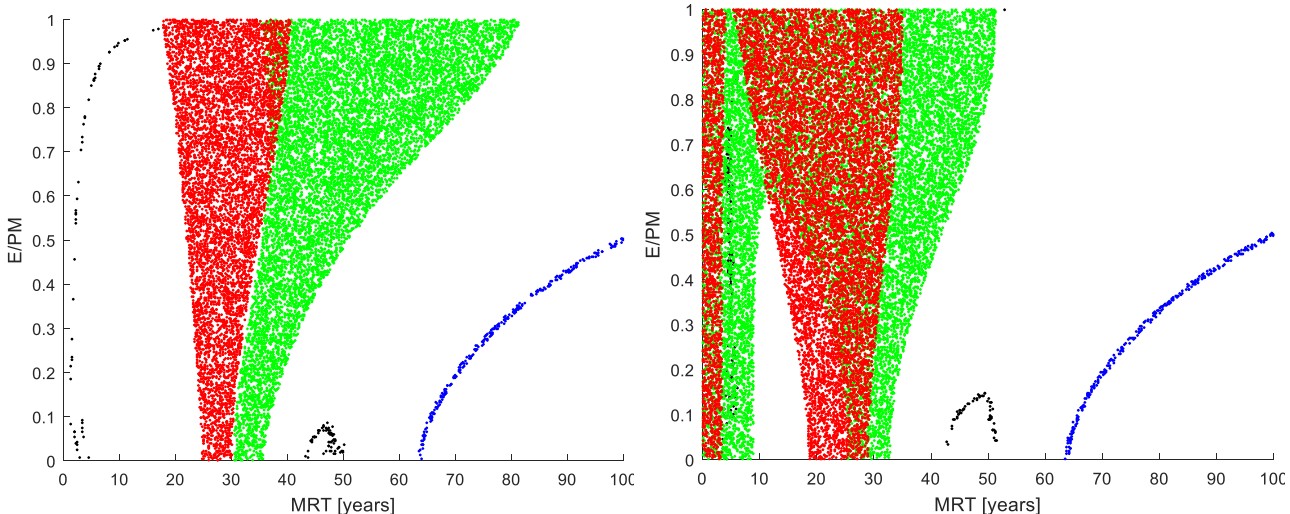

5  **Figure 8: Typical EPM parameter could be inferred with tritium (black), CFC-11 (green), CFC-12 (red) and CFC-113 (blue). In both figures tritium and CFC-11 and CFC-12 inferred LPM parameter clouds overlap, i.e. age interpretations agree; in both figures, tritium gives ambiguous age estimates, in right figure CFC-11 and CFC-12 also give ambiguous age estimates due to recently falling atmospheric concentrations.**

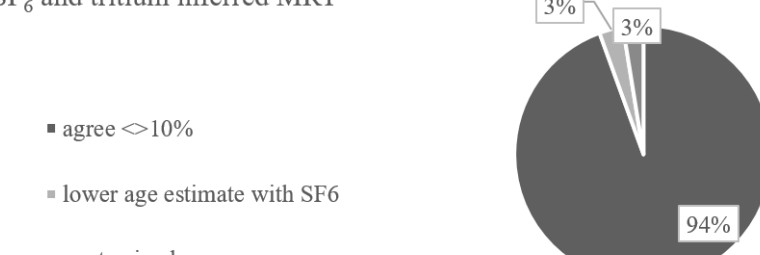

**Figure 9: Agreement of SF₆ and tritium inferred MRTs. The data suggest that for most sites SF₆ inferred MRTs match tritium inferred MRTs. Only few sites (3 %) were contaminated with SF₆ and another 3 % showed lower age estimates with SF₆ than with tritium, which could result from a thick unsaturated zone and associated travel time to the aquifer.**





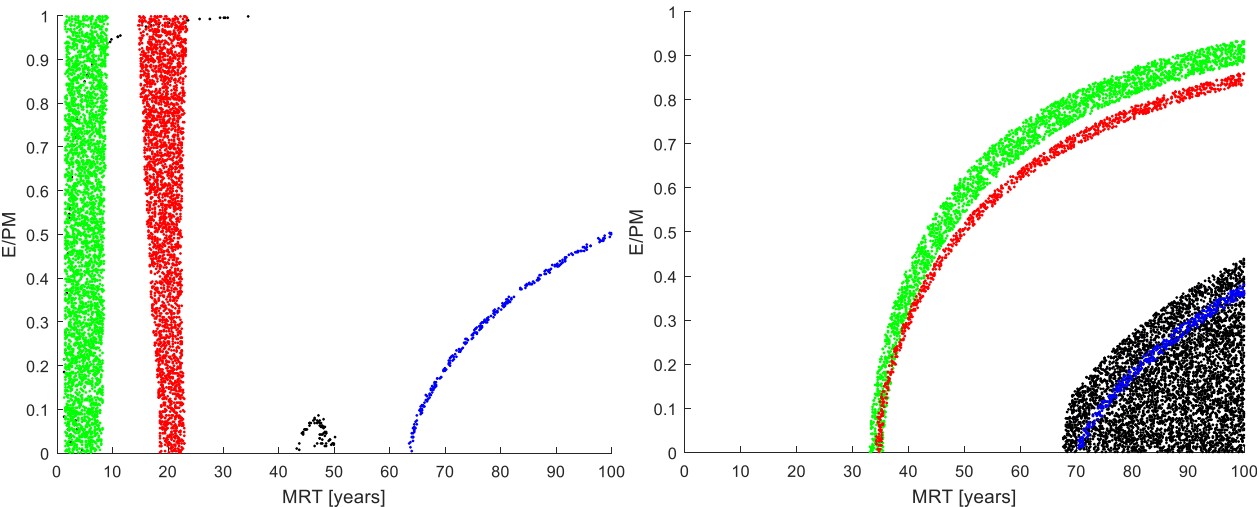

**Figure 10: Typical EPM parameter could be inferred with tritium (black), SF$_6$ (green), Halon-1301 (red) and CFC-113 (blue). In both figures some agreement of age interpretations is evident through overlapping LPM parameter clouds. In the left figure tritium age interpretation is ambiguous; the younger part agrees with the SF$_6$ and Halon-1301 inferred age. In the right figure the tritium and CFC-113 inferred age agrees, but differs from Halon-1301 and SF$_6$ inferred ages (which agree with each other).**





agreement of tritium and Halon-1301 inferred MRTs

■ agree <>10%

■ 10-25% higher Halon-1301 MRT

■ 25-50% higher Halon-1301 MRT

■ 50-100% higher Halon-1301 MRT

■ >100% higher Halon-1301 MRT

■ contamined > 25%

agreement of tritium and Halon-1301 inferred E/PMs

■ agree <>10%

■ 10-25% higher Halon-1301 MRT

■ 25-50% higher Halon-1301 MRT

■ 50-100% higher Halon-1301 MRT

■ >100% higher Halon-1301 MRT

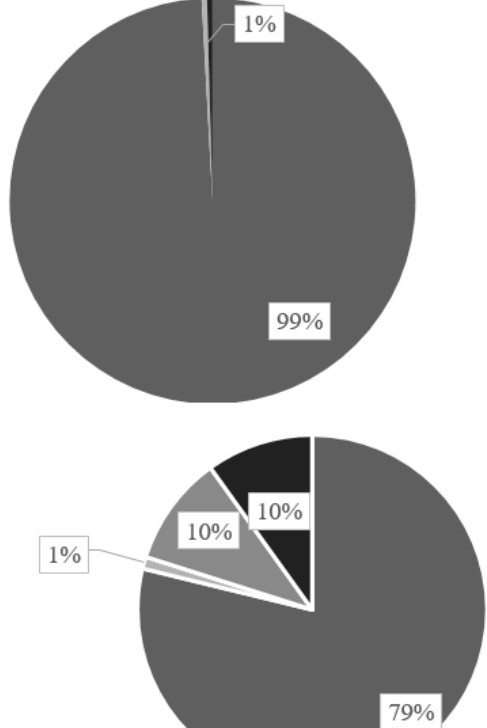

**Figure 11: Summary of performance of Halon-1301 as an age tracer compared to tritium in predicting the MRT (upper) and the mixing parameter E/PM (lower). Overall, at 99 % of the sites MRTs agree within +/- 10 %.**

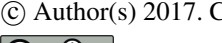


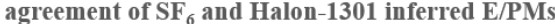

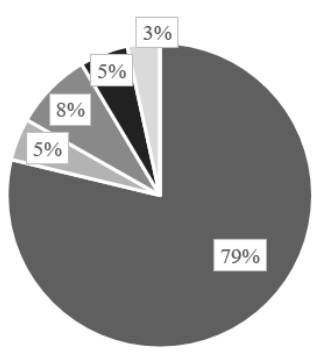

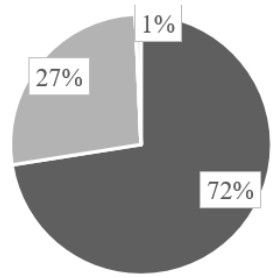

**Figure 12: Summary of performance of Halon-1301 as an age tracer compared to SF6 in predicting the MRT (upper) and the mixing parameter E/PM (lower). Overall, at 79 % of the sites MRTs agree within +/- 10 %.**

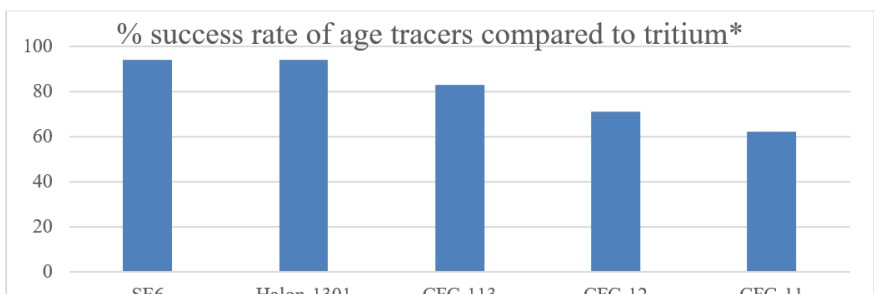

**Figure 13: Success rate of Halon-1301, SF₆ and the CFC -11 and -12 compared to tritium in this study; *assuming tritium has a success rate of 100 %**





**Appendix A**

**Figure 14: Behavioural age information (MRT and E/PM) inferred from SF$_6$ (red), tritium (black) and Halon-1301 (blue) for sites 1 to 45**





**Figure 15: Behavioural age information (MRT and E/PM) inferred from SF$_6$ (red), tritium (black) and Halon-1301 (blue) for sites 46 to 90**





Figure 16: Behavioural age information (MRT and E/PM) inferred from SF$_6$ (red), tritium (black) and Halon-1301 (blue) for sites 91 to 133





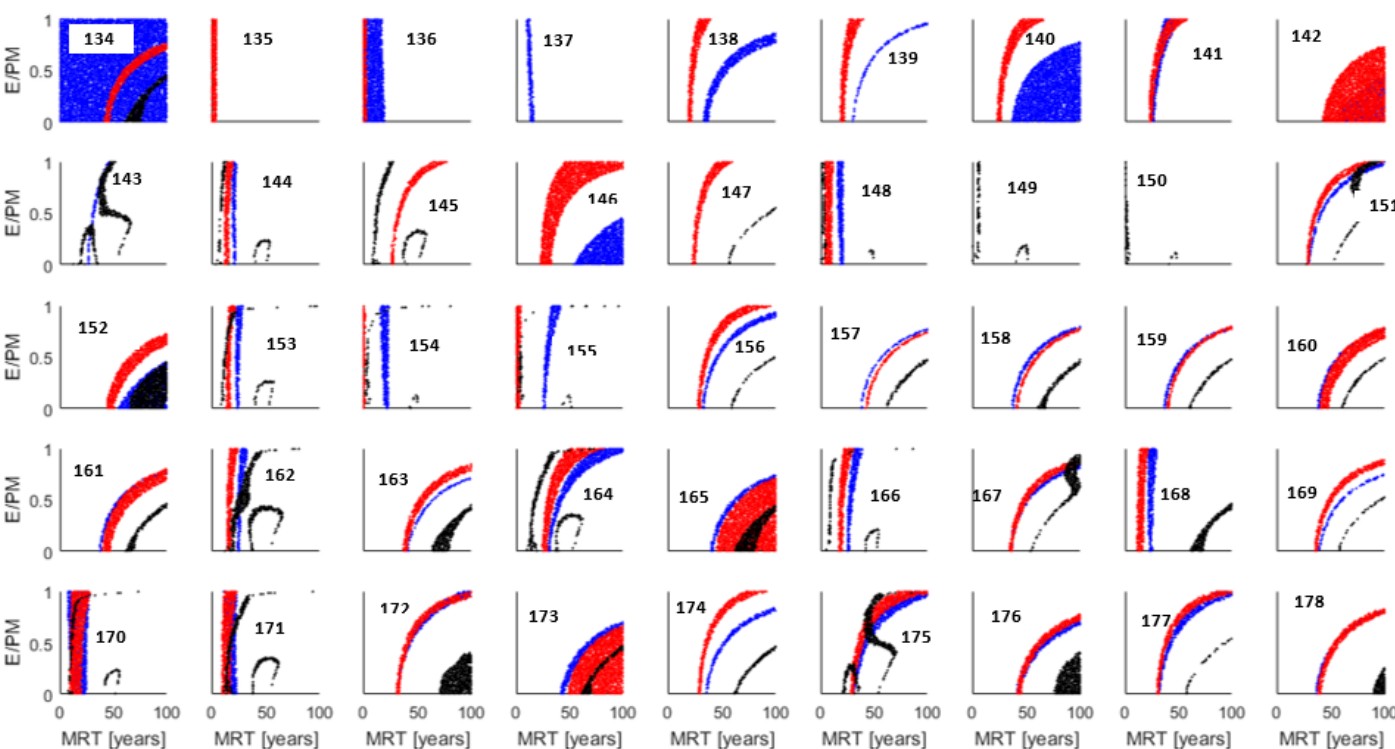

Figure 17: Behavioural age information (MRT and E/PM) inferred from SF$_6$ (red), tritium (black) and Halon-1301 (blue) for sites 134 to 178



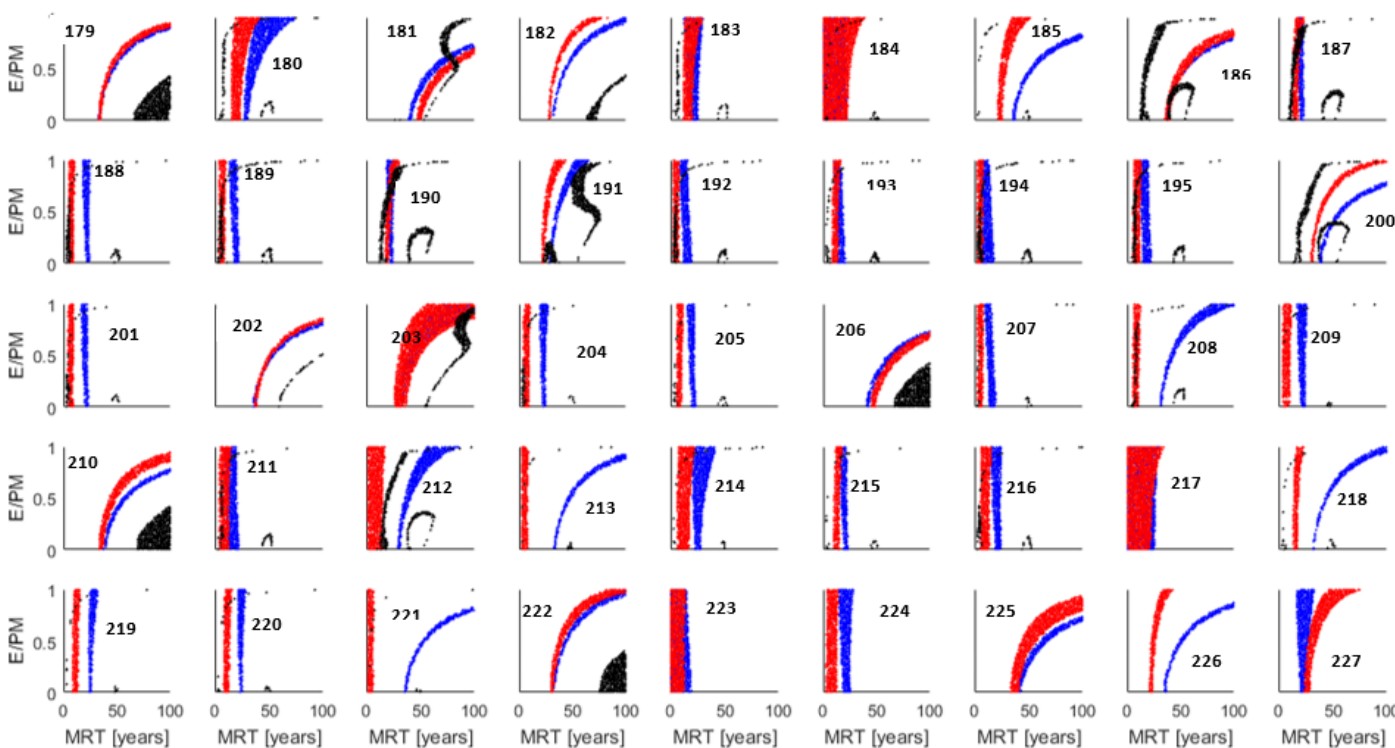

**Figure 18: Behavioural age information (MRT and E/PM) inferred from SF$_6$ (red), tritium (black) and Halon-1301 (blue) for sites 179 to 227**





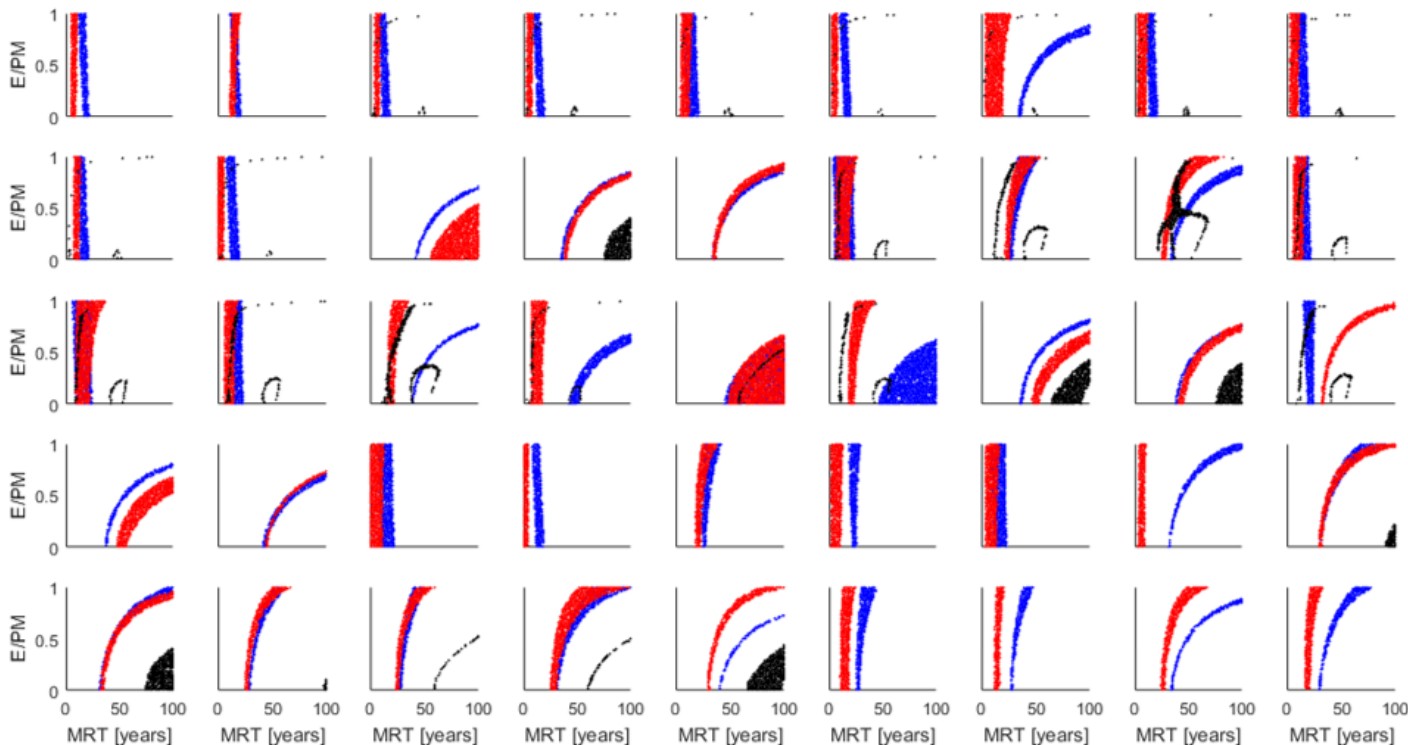

**Figure 19: Behavioural age information (MRT and E/PM) inferred from SF$_6$ (red), tritium (black) and Halon-1301 (blue) for sites 228 to 270**




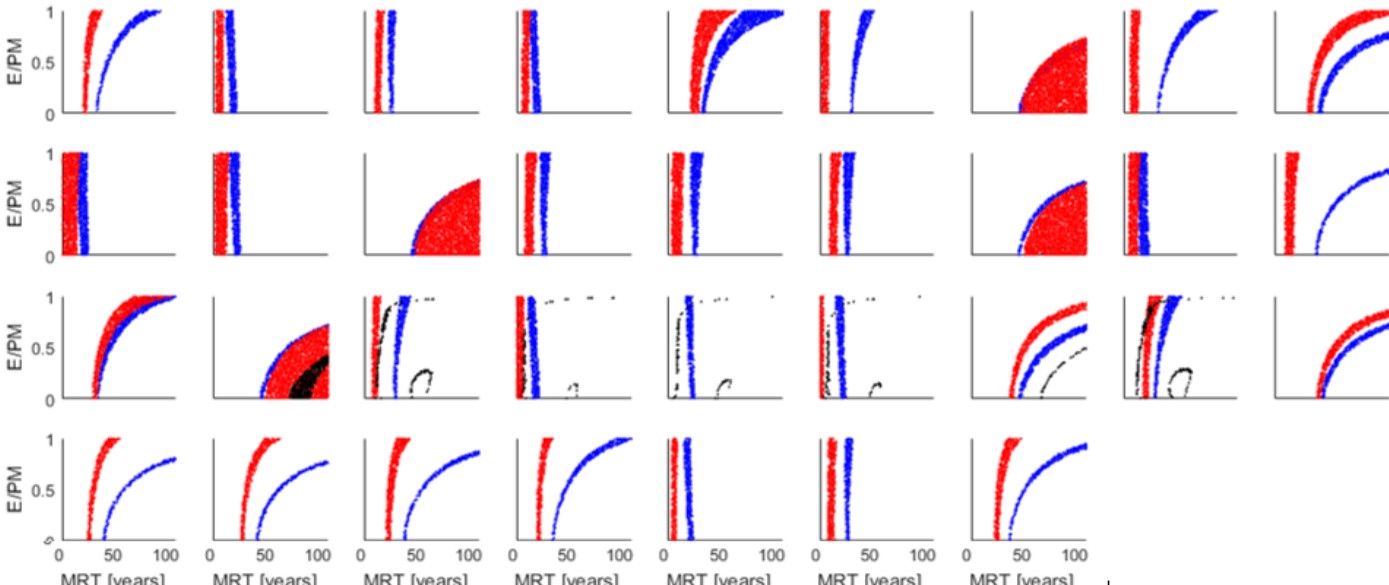

**Figure 20: Behavioural age information (MRT and E/PM) inferred from SF$_6$ (red), tritium (black) and Halon-1301 (blue) for sites 268 to 302**