# Peer review of "Halon-1301 – further evidence of its performance as an age tracer in New Zealand groundwater"

_Hydrology and Earth System Sciences, 2017_

## Referee Comment (RC1) · D. Gooddy (Referee) · 30 Mar 2017

Review of Halon 1301 – Further evidence of its performance as an age tracer in New Zealand Groundwater This manuscript presents and extensive data set for H1301, building on that previously published by the lead author in HESS 19, 2775-2789, 2015 and WRR, 50, WR015818, 2014.

Main Comments:

Despite the impressive data set I was slightly disappointed at the level of greater understanding that was gained from this. In particular, the unavailability of other tracers at some sites which would have hopefully given greater lucidity as to the retardation/removal processes taking place. I think this is a significant weakness in the paper although not one that the authors can rectify. I do think however that more thought needs to go into the discussion as this is key to the main knowledge advancement that the paper could potentially provide.

Where low concentrations of H1301 are found, have the authors considered degassing of N2 (as a result of denitrification) or CH4 as possible mechanisms for removal. Without any NO3 data this is hard for the reviewer to assess. I would therefore refer the authors to Visser et al 2007 (WRR 43, 10 W10434) and Visser et al. 2009 (JoH 369, 4-4, 427-439) where the issue of tracer degassing is discussed in extensive detail.

Related to this, I am interested that the authors are using N2/Ar ratios to correct for excess air, rather than the more normally accepted Ne. Could they comment on the possible issues relating to this, especially if denitrification is taking place.

As a general observation there are far too many figures and figures within figures – As these are not really discussed in any detail, the true significance is not clear.

Minor Comments:

P1 Line 13. Could not rather than "couldn't"

P1 Line 29. More description on the (speculation?) causes of H1301 reduction is needed here. In the Introduction section you are only really referring to recent groundwater age indicators and you need to be explicit about that. As a general over view of the state of the art I would refer the authors to Aqulina et al 2014 (Applied Geochemistry 50, 115-117) and Darling et al 2012 (Applied Geochemistry 27, 9, 1688-1697).

P2 line 13 would add Darling and Gooddy 2007 (Science of the Total Environment 387, 353-362)

P2 Line 15 after ambiguous age interpretations add Suckow 2014 (Applied Geochemistry 50, 222-230)

P3 Line 12 replace 'They' with Bartyzel et al (2016)

P4 Line 8 add reference to Oster 1996

P6 Question. Is the input function for S Hemisphere and N hemisphere the same for H1301? Some reference to the differences would be helpful for other/future practitioners.

P8 Line 19. Delete 'in fact'

P9 Line 18. Give reference for the 'issues' eluded to.

P11 Line 10. Need to justify assertion that T is one of the 'most reliable'. What do you mean by reliable?

P11 Line 14. The input of SF6 is exponential and not 'near linear'.

P13 line 15. Add in text relating to degassing potential.

---

## Referee Comment (RC2) · A. Suckow (Referee) · 28 May 2017

A. Suckow (Referee)

axel.suckow@csiro.au

**General Comments**

This manuscript massively extents the data set for Halon 1301 in New Zealand (>300 measurements) compared to the 2015 paper in HESS of the same authors, and compares its usability with tritium, $SF_6$ and the CFCs (CFC-11, CFC-12, CFC-113). The analysis uses a lumped parameter (LPM) approach – the exponential piston flow model (EPM) with an evaluation of mean residence time (MRT) and volume ratio (E/PM) for this model – to assess agreement in "groundwater age" as inferred by the different tracers. It is very well presented and besides the obvious demonstration of the usefulness of H1301 the article also shows some valuable new approaches in demonstrating and

quantifying agreement in lumped parameter model results. It is clearly worth publishing with minor revisions in HESS.

**Specific Comments**

Although using only one specific shape of an age distribution (the EPM), the paper does a good job in evaluating comparability of LPM results for different tracers for the same water sample as "agreement in inferred age". Of special value here is the 2d-plot of the E/PM parameter versus the MRT for different tracers if the model results are in the 1-sigma range of the measurement. This is a very useful way to display these results that I have not seen before. The developed metric, however, is worth discussing in more detail.

While I completely agree not to use MRT only, I have a problem with the attempted metric. No overlap of the two data clouds in Fig. 4 simply means that the two tracers give differing results which cannot be brought into a 1-sigma agreement with any parameter combination. So what is the meaning of the "Euclidian distance between two data clouds"? And what is the "% difference in MRT and mixing parameter inferred with two tracers"? Is the percentage (distance divided by what?) taken from the MRT and mixing parameter of the two nearest points or from the whole axis? For example, if the nearest two point in figure 4 left have 20 and 30 years MRT, is the percentage in MRT then (30-20)/(20)=50%? Or is it (30-20)/100=10%?

Most of the following paper uses a 10% criterion on this distance as "agreement" (Fig. 6, 7, 9, 11, 12) which is misleading, because any percentage >0 means the two results actually disagree. Perhaps a better way for quantifying agreement or disagreement would be to use a 1-sigma and a 2-sigma evaluation. Overlap of the clouds generated with 1 sigma would be good agreement, overlap of clouds with 2 sigma still agreement with a certain smaller probability. No overlap of data clouds generated with 3 sigma would be clear disagreement.

In case of disagreement (e.g. 6% of the sites with $SF_6$ and $^3H$ available) an evaluation of the uncertainty in recharge temperature, recharge altitude and excess air may be valuable – perhaps this would bring the results into agreement within the 1 sigma uncertainty of these parameters? Similar for H1301, since its dependency on temperature, altitude and excess air is different than for $SF_6$.

**Technical Corrections**

Attempting to determine a MRT of 150 years with the given tracers (P. 3 line 22) is too ambitious and does not acknowledge the high quality of LPM presentation of the rest of the paper. None of the discussed tracers is sensitive to water recharged prior to 1950 (not even with the high sensitivity reached by Uwe for tritium). This is 67 years, not 150. Even using MRT instead of "age": an EPM with an E/PM of 0.1 (bottom row in plots of figure 15-20) and MRT of 75 years contains none of the tracers (all water in it is >67.5years old), and only <0.001 parts of water older than 120years. There is a good reason why all plots in Fig. 4, 8, 10, 15-20 have MRTs only until 100 years.

P.5 L16: Salinity may be negligible for groundwater in NZ, not so in other parts of the world (e.g. Australia).

In Figure 1 using the same colours for the symbols as in the pie chart would increase readability.

Figures 6, 7, 9, 11, 12 use "agree <> 10%". This is a strange use of the symbols "<>". I think a better way to express what you mean is "disagreement <10%", see discussion above.

Figures 6, 7, 9, 11, 12 display the word "contamined" which I assume has to be "contaminated".

Figures 6, 7, 9, 11  12 would be improved by having colour in the pie chart and legend.

The low-MRT branch of tritium is invisible in Fig. 8 right and Fig. 10 left. Put that on top of the other plots.

An explanation is needed why in Figure 15-20 the displays 53, 56, 57, 63-68, 134 show the whole area as blue. Uncertainty of H1301 measurement too large?

---

## Author Comment (AC1) · 22 Jun 2017

Dear Daren, We greatly appreciate your valuable feedback on our manuscript. Below, we detail our responses to your comments. We intend to closely follow all of your suggestions.

Comments from D. Gooddy (Referee) dcg@bgs.ac.uk This manuscript presents and extensive data set for H1301, building on that previously published by the lead author in HESS 19, 2775-2789, 2015 and WRR, 50, WR015818, 2014. Main Comments: Despite the impressive data set I was slightly disappointed at the level of greater understanding that was gained from this. In partic-

ular, the unavailability of other tracers at some sites which would have hopefully given greater lucidity as to the retardation/removal processes taking place. I think this is a significant weakness in the paper although not one that the authors can rectify.

–> Thanks for pointing this out. We agree with your comment and intend to tone down our message re further insight into the causes of reduced Halon-1301 concentrations in abstract and introduction.

I do think however that more thought needs to go into the discussion as this is key to the main knowledge advancement that the paper could potentially provide.

–> Thanks, we agree with your comment and intend to include a more detailed discussion around degassing following your suggestions below.

Where low concentrations of H1301 are found, have the authors considered degassing of N2 (as a result of denitrification) or CH4 as possible mechanisms for removal. Without any NO3 data this is hard for the reviewer to assess. I would therefore refer the authors to Visser et al 2007 (WRR 43, 10 W10434) and Visser et al. 2009 (JoH 369, 4-4, 427-439) where the issue of tracer degassing is discussed in extensive detail.

–> Thanks for pointing this out. Unfortunately we do not have NO3 data for over half of the sites to further assess degassing as possible cause of reduced Halon-1301 concentrations. Further, we note that degassing would affect all gaseous tracers (including Halon-1301) – with the most impact to the least soluble tracer, making it even harder to properly assess degassing as possible cause. However, we note that we assess the oxygen content of the sample (i.e. whether the sample is oxic or anoxic), which should give a good indication of whether deN could have taken place – and most of the samples that indicated reduced Halon-1301 concentrations were indeed anoxic.

–> Following your suggestions, we intend to discuss degassing as possible cause of reduced Halon-1301 concentrations (see also comment re Ne/Ar and determination of excess N in as response to your next comment). –> Further, to highlight that we

had previously considered a more comprehensive range of possible causes of Halon-1301 'removal' (in our HESS 19, 2775-2789, 2015 paper), we intend to include a more detailed summary of these in our manuscript.

Related to this, I am interested that the authors are using N2/Ar ratios to correct for excess air, rather than the more normally accepted Ne. Could they comment on the possible issues relating to this, especially if denitrification is taking place.

–> While Ne/Ar is more robust, N2/Ar is much simpler to measure and still provides a useful excess air and recharge temperature correction in most cases. Significant denitrification (excess N) can be identified by anomalously high recharge temperatures. In such cases, excess N is corrected for by applying the mean annual air temperature. That method also allows for estimation of excess N. We think further assessment of degassing into excess N as possible cause of Halon-1301 'removal' would add significantly to the discussion on reduced Halon-1301 concentrations. We therefore intend to address it using the estimated excess N and available NO3 and CH4 data.

As a general observation there are far too many figures and figures within figures – As these are not really discussed in any detail, the true significance is not clear.

–> Thanks for your comment. We intend to assess the significance of each figure and remove them if we feel is needed.

Minor Comments: P1 Line 13. Could not rather than "couldn't" P1 Line 29. More description on the (speculation?) causes of H1301 reduction is needed here. In the Introduction section you are only really referring to recent groundwater age indicators and you need to be explicit about that. As a general over view of the state of the art I would refer the authors to Aqulina et al 2014 (Applied Geochemistry 50, 115-117) and Darling et al 2012 (Applied Geochemistry 27, 9, 1688-1697). P2 line 13 would add Darling and Gooddy 2007 (Science of the Total Environment 387, 353-362) P2 Line 15 after ambiguous age interpretations add Suckow 2014 (Applied Geochemistry 50, 222-230) P3 Line 12 replace 'They' with Bartyzel et al (2016) P4 Line 8 add reference to

Oster 1996 P6 Question. Is the input function for S Hemisphere and N hemisphere the same for H1301? Some reference to the differences would be helpful for other/future practitioners. P8 Line 19. Delete 'in fact' P9 Line 18. Give reference for the 'issues' eluded to. P11 Line 10. Need to justify assertion that T is one of the 'most reliable'. What do you mean by reliable? P11 Line 14. The input of SF6 is exponential and not 'near linear'. P13 line 15. Add in text relating to degassing potential.

–> Thanks for the above comments. We intend to follow your suggestions and make changes as per above comments.

–––––––––––––––––––––––––––

---

## Author Response (AR1)

Dear editor,

Many thanks for giving us the opportunity to comment on the reviewers' feedback.

Both reviewers had only minor comments. We followed all of their suggestions as detailed below.

Dear Daren,

We greatly appreciate your valuable feedback on our manuscript. Below, we detail our responses to your comments. We closely followed all of your suggestions.

**Comments from D. Gooddy (Referee) dcg@bgs.ac.uk

This manuscript presents and extensive data set for H1301, building on that previously published by the lead author in HESS 19, 2775-2789, 2015 and WRR, 50, WR015818, 2014.

**Main Comments:** Despite the impressive data set I was slightly disappointed at the level of greater understanding that was gained from this. In particular, the unavailability of other tracers at some sites which would have hopefully given greater lucidity as to the retardation/removal processes taking place. I think this is a significant weakness in the paper although not one that the authors can rectify.

➔ Thanks for pointing this out. We agree with your comment toned down our message re further insight into the causes of reduced Halon-1301 concentrations in abstract and conclusion.

I do think however that more thought needs to go into the discussion as this is key to the main knowledge advancement that the paper could potentially provide.

➔ Thanks, we agree with your comment and included a more detailed discussion around degassing following your suggestions below.

Where low concentrations of H1301 are found, have the authors considered degassing of N2 (as a result of denitrification) or CH4 as possible mechanisms for removal. Without any NO3 data this is hard for the reviewer to assess. I would therefore refer the authors to Visser et al 2007 (WRR 43, 10 W10434) and Visser et al. 2009 (JoH 369, 4-4, 427-439) where the issue of tracer degassing is discussed in extensive detail.

➔ Thanks for pointing this out. Unfortunately we do not have NO3 data for over half of the sites to further assess degassing as possible cause of reduced Halon-1301 concentrations. However, following your suggestions we determined excess N as described below and discuss degassing as possible cause of reduced Halon-1301 concentrations.

➔ Further, to highlight that we had previously considered a more comprehensive range of possible causes of Halon-1301 'removal' (in our HESS 19, 2775-2789, 2015 paper), we included a more detailed summary of these in the introduction (page 3, line 14ff.).

Related to this, I am interested that the authors are using N2/Ar ratios to correct for excess air, rather than the more normally accepted Ne. Could they comment on the possible issues relating to this, especially if denitrification is taking place.

➔ While Ne/Ar is more robust, N2/Ar is much simpler to measure and still provides a useful excess air and recharge temperature correction in most cases. Significant denitrification (excess N) can be identified by anomalously high recharge temperatures. In such cases, excess N is corrected for by applying the mean annual air temperature. That method also allows for estimation of excess N. We think further assessment of degassing into excess N as possible cause of Halon-1301 'removal' would add significantly to the discussion on reduced Halon-1301 concentrations. We therefore addressed the above using the estimated presence of excess N. Please see page 5, line 5 ff. for a description of the method and page 13, line 20ff for results.

As a general observation there are far too many figures and figures within figures – As these are not really discussed in any detail, the true significance is not clear.

➔ Thanks for your comment. We assessed the significance of each figure and removed figure 6 and 7.

**Minor Comments**:

P1 Line 13. Could not rather than "couldn't" ➔ Agreed and changed accordingly.

P1 Line 29. More description on the (speculation?) causes of H1301 reduction is needed here. ➔ Yes, please see changes on page 3, line 14ff.

In the Introduction section you are only really referring to recent groundwater age indicators and you need to be explicit about that. As a general over view of the state of the art I would refer the authors to Aqulina et al 2014 (Applied Geochemistry 50, 115-117) and Darling et al 2012 (Applied Geochemistry 27, 9, 1688-1697). P2 line 13 would add Darling and Gooddy 2007 (Science of the Total Environment 387, 353-362) ➔ Agreed. Please see changes in page 2, line 18 and 19.

P2 Line 15 after ambiguous age interpretations add Suckow 2014 (Applied Geochemistry 50, 222-230)  ; P3 Line 12 replace 'They' with Bartyzel et al (2016); P4 Line 8 add reference to Oster 1996 ➔ Thanks, changed accordingly.

P6 Question. Is the input function for S Hemisphere and N hemisphere the same for H1301? Some reference to the differences would be helpful for other/future practitioners. ➔ We added a note to the caption of figure 1 to clarify the above.

P8 Line 19. Delete 'in fact'; P9 Line 18. Give reference for the 'issues' eluded to. ➔ Agreed and changes made accordingly.

P11 Line 10. Need to justify assertion that T is one of the 'most reliable'. What do you mean by reliable? ➔ Limitations of the tracers were described in the introduction. We included a link to the introduction to clarify the above.

P11 Line 14. The input of SF6 is exponential and not 'near linear'. ➔ Clarified: near linear from 1985 onwards.

P13 line 15. Add in text relating to degassing potential. ➔ Thanks, done accordingly.

Dear Axel,

Many thanks for your valuable feedback on our manuscript. Below, we detail our responses to your comments. We closely followed your suggestions.

**Comments from A. Suckow (Referee) axel.suckow@csiro.au

**General Comments:** This manuscript massively extents the data set for Halon 1301 in New Zealand (>300 measurements) compared to the 2015 paper in HESS of the same authors, and compares its usability with tritium, SF6 and the CFCs (CFC-11, CFC-12, CFC-113). The analysis uses a lumped parameter (LPM) approach – the exponential piston flow model (EPM) with an evaluation of mean residence time (MRT) and volume ratio (E/PM) for this model – to assess agreement in "groundwater age" as inferred by the different tracers. It is very well presented and besides the obvious demonstration of the usefulness of H1301 the article also shows some valuable new approaches in demonstrating and quantifying agreement in lumped parameter model results. It is clearly worth publishing with minor revisions in HESS.

**Specific Comments:** Although using only one specific shape of an age distribution (the EPM), the paper does a good job in evaluating comparability of LPM results for different tracers for the same water sample as "agreement in inferred age". Of special value here is the 2d-plot of the E/PM parameter versus the MRT for different tracers if the model results are in the 1-sigma range of the measurement. This is a very useful way to display these results that I have not seen before. The developed metric, however, is worth discussing in more detail.

While I completely agree not to use MRT only, I have a problem with the attempted metric. No overlap of the two data clouds in Fig. 4 simply means that the two tracers give differing results which cannot be brought into a 1-sigma agreement with any parameter combination. So what is the meaning of the "Euclidian distance between two data clouds"? And what is the "% difference in MRT and mixing parameter inferred with two tracers"? Is the percentage (distance divided by what?) taken from the MRT and mixing parameter of the two nearest points or from the whole axis? For example, if the nearest two point in figure 4 left have 20 and 30 years MRT, is the percentage in MRT then (30-20)/(20)=50%? Or is it (30-20)/100=10%?

> ➔ Thanks for pointing this out. We agree that more discussion around the developed metric would be great. We included your comment re "No overlap of the inferred LPM parameter clouds implies that the two tracers give differing results which cannot be brought into a 1-sigma agreement with any parameter combination."(page 8, line 25ff). We also included an equation for the metric '% difference in MRT' to clarify its definition. ➔ see Eqn. 5

Most of the following paper uses a 10% criterion on this distance as "agreement" (Fig. 6, 7, 9, 11, 12) which is misleading, because any percentage >0 means the two results actually disagree. Perhaps a better way for quantifying agreement or disagreement would be to use a 1-sigma and a 2-sigma evaluation. Overlap of the clouds generated with 1 sigma would be good agreement, overlap of clouds with 2 sigma still agreement with a certain smaller probability. No overlap of data clouds generated with 3 sigma would be clear disagreement. In case of disagreement (e.g. 6% of the sites with SF6 and 3H available) an evaluation of the uncertainty in recharge temperature, recharge altitude and excess air may be valuable – perhaps this would bring the results into agreement within the 1 sigma uncertainty of these parameters? Similar for H1301, since its dependency on temperature, altitude and excess air is different than for SF6.

➔ Thanks for your suggestions. We generally agree that it is a good idea to use multiple distance measure levels to assess agreement/disagreement. As this is one of the first attempts to use that type of approach, we decided to use 1 sigma uncertainty for the following reason. As we illustrate, most inferred LPM parameter clouds are already relatively large when using the 1 sigma criterion implying large uncertainties in inferred age parameters (this is not surprising as we only use 1 measurement to infer age). We are worried that when using a 2 sigma (or higher) distance criterion, huge inferred LPM parameter may suggest to the reader that the tracers are of no use. We added the 10% criterion to add another level of check assessing relative distance in addition to absolute distance of the cloud as per 1 sigma criterion.

➔ In general, a variety of objective functions (which our criterion in principle is), is increasingly used in the hydrological modelling community suggesting that there is no one criterion that should be applied everywhere. We included a note on the relative novelty of this approach and highlight that its general applicability needs to be assessed further for other datasets. – see page 9 ff.

**Technical Corrections**

Attempting to determine a MRT of 150 years with the given tracers (P. 3 line 22) is too ambitious and does not acknowledge the high quality of LPM presentation of the rest of the paper. None of the discussed tracers is sensitive to water recharged prior to 1950 (not even with the high sensitivity reached by Uwe for tritium). This is 67 years, not 150. Even using MRT instead of "age": an EPM with an E/PM of 0.1 (bottom row in plots of figure 15-20) and MRT of 75 years contains none of the tracers (all water in it is >67.5years old), and only <> 10%". This is a strange use of the symbols "<>". I think a better way to express what you mean is "disagreement

➔ Thanks for the above comment. To clarify, a water with MRT 75y and E/PM 0.1 contains 59% of water younger than 67.5 years. Water with MRT 150 years still contains 33% of water younger than 67.5 years and therefore contains significant amounts of tritium. For this model example we can detect up to 250 years MRT. So by saying 150 years we are not over-ambitious. However, we agree that for such old (nearly tritium-free) waters, the aggregation error is large – the old component of the water could be thousands of years old and therefore the true mean age be significantly older [see e.g. Stewart et al., 2016 - doi:10.5194/hess-2016-532]. We included a footnote on page 4 to explain the above more clearly.

[revised manuscript text omitted]

10   and $N_2$ determination described in Heaton and Vogel (1981). We note that that method is sensitive to excess $N_2$ produced e.g. by denitrification. We determined the presence of excess $N_2$ (as a product of denitrification) on the basis of anomalously high inferred recharge temperatures determined from $N_2$/Ar ratios. Specifically, we identified samples for which the inferred recharge temperature were significantly higher than the mean annual air temperature at that location (using climate data from NIWA (2016)). Assessment of the performance of Halon-1301 as a groundwater age tracer at these sites allows for assessment

15   of degassing into excess $N_2$ as possible cause of reduced Halon-1301 concentrations.

**TABLE 1 here**

[revised manuscript text omitted]
). No overlap of the inferred LPM parameter clouds implies that the two tracers give differing results which cannot be brought into a 1-sigma agreement with any parameter combination. If they were non-overlapping (i.e. different), we determined the shortest distance of inferred LPM parameter populations as a measure of difference. As a measure of the shortest distance, we determined the nearest neighbour and minimum Euclidian distance between two data clouds in Matlab software (Muja and Lowe, 2009). From that, the % difference in MRT and mixing parameter inferred with two tracers (e.g. $SF_6$ and Halon-1301) was determined as:-

$$\Delta\varepsilon\%(MRT) = \frac{d_{\min\,tracer1,tracer2}}{\text{mean}(MRT_{tracer1}, MRT_{tracer2})} * 100 \tag{5}.$$

We note that a variety of objective or fitting functions is increasingly used in the hydrological modelling community suggesting that there is no one criterion that should be applied everywhere (e.g. Beven and Binley, 2014). The general applicability of the approach suggested here (using 1 sigma and 10% distance criterion) needs to be assessed further on other datasets.

[revised manuscript text omitted]

Discrepancy of recharge temperature to mean annual air temperature of the region suggesting excess $N_2$ was found in 29 sites. Of these 25 showed matching SF6, Halon-1301, CFC-12, CFC-113 and tritium inferred ages suggesting that degassing into headspace created by denitrification did not affect any of the gaseous tracers at these sites. At the remaining four sites, Halon-1301 inferred MRTs were significantly different to CFC-113 and $SF_6$ inferred MRTs. However, at these four sites Halon-1301 inferred MRTs matched those inferred from CFC-12 (tritium was unavailable at 2 of these sites, at 1 site Halon-1301 inferred MRT and was different to that inferred with tritium, and at the remaining site Halon-1301 inferred MRT matched that inferred from tritium). This suggests that degassing did effect the gaseous tracers, the most least soluble ones ($SF_6$ and CFC-113). Halon-1301 (and similarly CFC-12) appear to be less effected by degassing into headspace created by de-nitrification, production of methane or when groundwater is brought to the ground surface, making it a more reliable age tracer than $SF_6$ and CFC-113 in these environments.

**TABLE 4 here**

In summary, our findings suggest that Halon-1301 performed well as an age tracer at the majority of groundwater sites. Although reduced Halon-1301 concentrations were found in a few samples, resulting in misleading Halon-1301 inferred age estimates, overall Halon-1301 performed significantly better than the CFCs, which are prone to degradation and contamination (shown in this and our previous study). Figure 11 summarizes the performance of the tracers used in this study, highlighting that Halon-1301 performs almost as well as $SF_6$ and tritium, and with a much higher success rate than the CFCs in this study. In particular, Halon-1301 is significantly more reliable than the CFCs (which were either degraded or contaminated at as many as 30 % of the sites) and in some cases $SF_6$ (for six contaminated $SF_6$ samples, Halon-1301 still matched age estimates from tritium).

**FIGURE 11 here**

**4 Conclusion**

In summary, this study presented an extensive assessment of the performance of Halon-1301 in 302 groundwater samples across New Zealand. We showed that Halon-1301 had a high reliability as an age tracer, similar to that of $SF_6$ and tritium. It performed much better than the CFCs, CFC-11 and -12, which are prone to degradation and contamination. Both degradation

and contamination lead to non-conforming age estimates. For example, despite some groundwater samples showing evidence of contamination from industrial or agricultural sources (inferred by elevated CFC concentrations), no sample showed significantly elevated concentration of Halon-1301, which suggests there were no local anthropogenic or geologic sources of Halon-1301 contamination.

5    Like any other tracer, the use of Halon-1301 as a groundwater age tracer has its limitations. In this and our previous study, reduced concentrations of Halon-1301 were found. Causes for these are likely degradation,  sorption and/or degassing into headspace or excess $N_2$. Halon-1301 appears to be less affected by degassing  than $SF_6$ and CFC-113 due to  its higher solubility. Although we provided further evidence for degradation being the main reason for reduced Halon-1301 concentrations, we could not fully determine the reasons for the reduced concentrations. We hope that future

10   studies will explore this matter further. Knowing the cause of reduced Halon-1301 concentrations is important as it can help predict its reliability as an age tracer in different groundwater environments.

Further study is also needed on time series Halon-1301 data to better understand how uncertainty in inferred age information can be constrained with multiple Halon-1301 data compared to other tracers, e.g. tritium and $SF_6$. In addition, the solubility of Halon-1301 needs to be better estimated to reduce uncertainty in the determination of Halon-1301 in groundwater and inferred

15   age.

Overall, we highly recommend the use of Halon-1301 as an age tracer, in particular its use in combination with $SF_6$. The simultaneous determination of Halon-1301 with $SF_6$ (and CFC-12) at no additional cost to sole $SF_6$ analysis, can reduce both tracer's limitations to ultimately obtain a more reliable inferred age than through the use of a single age tracer.

**Acknowledgments**

20   The New Zealand Ministry of Business and Innovation is thanked for funding in line with the Smart Aquifer Characterization (SAC) project.

[revised manuscript text omitted]

| Agreement of Halon-1301 inferred MRT with those inferred with CFCs and tritium | Redox state | excess $N_2$ | # of sites affected | Halon-1301 likely degraded? |
|---|---|---|---|---|
| Matching tritium and Halon-1301 inferred MRTs | Various | | 39 | No |
| Tritium and Halon-1301 MRTs do not match | Anoxic | Yes | 1 | **Yes, also possibly degassing into excess $N_2$** |
| Matching CFC-12, CFC-113 and Halon-1301 inferred MRTs, unavailable tritium data | Anoxic/unknown | No | 2 | No |
| Matching CFC-12 and Halon-1301 inferred MRTs, unavailable CFC-113 and tritium data | Various | Yes, 1 anoxic site | 5 | No |
| CFC-113 and Halon-1301 MRTs do not match, unavailable tritium and CFC-12 data | Oxic | No | 4 | No, Halon-1301 possibly retarded |
| | Anoxic | No | 1 | **Yes** |
| unavailable CFC and tritium data | Anoxic | Yes, 1 | 7 | **Cannot be excluded** |
| | Unknown | No | 2 | **Cannot be excluded** |
| | Oxic | No | 1 | **No** |

**FIGURES**

[Figure]

**Figure 1: Location of groundwater samples analysed for Halon-1301 in New Zealand. Groundwater was considered as oxic if the concentration of dissolved oxygen exceeded 0.5 mg/L and/or the concentration of dissolved iron and/or manganese was below 0.05 mg/L and methane was not present (and vice versa for anoxic water).**

[Figure]

**Figure 2: Southern hemisphere atmospheric concentrations of CFC-12, CFC-11, CFC-113, SF₆, Halon-1301 and tritium, using data from NOAA (available at ftp://ftp.cmdl.noaa.gov/hats) for the CFCs and SF₆; Morgenstern and Taylor (2009) for tritium. Concentrations of Halon-1301 in northern hemisphere are very similar to those in the southern hemisphere (see Figure 1 in Beyer et al., 2014 and references therein) suggesting Halon-1301 is well mixed in the atmosphere and can be applied as an age tracer in both hemispheres.**

[revised manuscript text omitted]